# ERK2 MAP kinase regulates SUFU binding by multisite phosphorylation of GLI1

A Jane Bardwell[1], Beibei Wu[2], Kavita Y Sarin[3] , Marian L Waterman[2] , Scott X Atwood[1] , Lee Bardwell[1]

**Crosstalk between the Hedgehog and MAPK signaling pathways occurs in several types of cancer and contributes to clinical resistance to Hedgehog pathway inhibitors. Here we show that MAP kinase-mediated phosphorylation weakens the binding of the GLI1 transcription factor to its negative regulator SUFU. ERK2 phosphorylates GLI1 on three evolutionarily conserved target sites (S102, S116, and S130) located near the high-affinity binding site for SUFU; these phosphorylations cooperate to weaken the affinity of GLI1–SUFU binding by over 25-fold. Phosphorylation of any one, or even any two, of the three sites does not result in the level of SUFU release seen when all three sites are phosphorylated. Tumor-derived mutations in R100 and S105, residues bordering S102, also diminish SUFU binding, collectively defining a novel evolutionarily conserved SUFU affinity–modulating region. In cultured mammalian cells, GLI1 variants containing phosphomimetic substitutions of S102, S116, and S130 displayed an increased ability to drive transcription. We conclude that multisite phosphorylation of GLI1 by ERK2 or other MAP kinases weakens GLI1-SUFU binding, thereby facilitating GLI1 activation and contributing to both physiological and pathological crosstalk.**

## Introduction

Integration between different signaling pathways is a crucial aspect of cellular regulation, as these pathways must work together in a coordinated manner to achieve specific physiological goals. On the other hand, such crosstalk signaling can also be dysregulated in disease, contributing significantly to pathology.

Indeed, there is evidence that crosstalk between the Hedgehog (HH) pathway and MAPK signaling pathways may be relevant to developmental signaling and to several types of cancer (reviewed in Aberger and Ruiz i Altaba [2014], Rovida and Stecca [2015], and Pietrobono et al [2019]). These cancers include basal cell carcinoma, the most frequent adult cancer; medulloblastoma, the most common childhood brain cancer; and melanoma and pancreatic adenocarcinoma, some of the deadliest of cancers (Teglund & Toftgard, 2010; Pak & Segal, 2016; Wu et al, 2017a; Raleigh & Reiter, 2019). Additional evidence suggests that MAPK crosstalk signaling may contribute to the emergence of clinical resistance to HH pathway inhibitors (Atwood et al, 2012; Boumahdi & de Sauvage, 2020).

In the developing embryo, HH signaling is widespread, and it plays a crucial role in patterning, proliferation and differentiation; in adults, HH signaling acts on stem/progenitor cells to regulate tissue homeostasis, regeneration and repair (Ingham et al, 2011; Robbins et al, 2012; Lee et al, 2016; Kong et al, 2019). HH, a secreted, cholesterol-modified ligand, binds to a transmembrane receptor designated Patched1 (PTCH1). Unliganded PTCH1 inhibits the function of the 7-transmembrane protein SMO, and this inhibition is relieved upon HH ligand binding to PTCH1. Interestingly, the initial steps of HH signaling occur in the primary cilium, an organelle that protrudes from the surface of most mammalian cells during growth arrest (Bangs & Anderson, 2017). Defective HH signaling results in birth defects, and dysregulated HH signaling is involved in a large number of cancers (Teglund & Toftgard, 2010; Pak & Segal, 2016; Wu et al, 2017a; Raleigh & Reiter, 2019).

The ultimate targets of activated SMO are the transcription factors GLI1, GLI2, and GLI3 (Hui & Angers, 2011). In unstimulated cells, the GLI proteins are found in cytoplasmic complexes with a 54-kD protein designated SUFU (named after the fly ortholog *Suppressor of fused*). Upon SMO activation, GLI-SUFU complexes have been observed to move into the primary cilium, wherein GLI proteins become activated by a mechanism that is thought to involve dissociation of SUFU, coupled with changes in the post-translational modification and processing of GLI (Humke et al, 2010; Tukachinsky et al, 2010; Niewiadomski et al, 2014). The active GLI factors then enter the nucleus and promote transcription. GLI proteins are DNA-binding zinc finger transcription factors that bind to the consensus element GACCACCCA. Active GLI1 has been shown to regulate target genes that include *CYCLIN D1* and *D2*, *FOXM1*, *BCL2*, *TERT*, and *GLI1* itself (Mazumdar et al, 2013).

SUFU is a key negative regulator of the three GLI transcription factors, and the dissociation of SUFU is considered to be a key step in GLI activation (Svard et al, 2006; Pak & Segal, 2016; Huang et al, 2018). A short, high-affinity binding site for SUFU, [120]SYGHLS[125], was

[1]Department of Developmental and Cell Biology, University of California, Irvine, CA, USA   [2]Department of Microbiology and Molecular Genetics, School of Medicine, University of California, Irvine, CA, USA   [3]Department of Dermatology, Stanford University School of Medicine, Stanford, CA, USA

Correspondence: bardwell@uci.edu

identified in the N-terminal portion of GLI1 (Dunaeva et al, 2003). Subsequent crystallographic studies showed that peptides containing these six residues bind in a narrow channel between the N-terminal and C-terminal lobes of the bi-lobed SUFU structure (Cherry et al, 2013; Zhang et al, 2013). Germline loss-of-function mutations in *SUFU* predispose to common pediatric and adult brain cancers (Taylor et al, 2002; Aavikko et al, 2012; Smith et al, 2014), and are also found in patients with Gorlin syndrome and Joubert syndrome (Pastorino et al, 2009; Smith et al, 2014; De Mori et al, 2017). Somatic *SUFU* mutations have also been found in various cancers, including sporadic basal cell carcinomas, where they have been shown to inappropriately enable GLI1-driven transcription (Urman et al, 2016). Loss of SUFU can also promote resistance to clinical HH-pathway inhibitors such as vismodegib (Sharpe et al, 2015; Zhao et al, 2015).

MAPK cascades are typically found embedded in signaling networks that transmit many different signals, including those initiated by growth and development factors, inflammatory stimuli, and cellular stresses (Raman et al, 2007; Morrison, 2012). A well-known example is the RAS/MAPK cascade, which transmits signals from growth factor receptors to effectors that promote cell division (Futran et al, 2013; Lavoie et al, 2020). Components of this pathway are frequently mutated in human cancer and developmental disorders, resulting in constitutive activation of the ERK1 and ERK2 MAPKs (Tidyman & Rauen, 2016; Bardwell, 2020). Activated MAPKs phosphorylate numerous targets, including many transcription factors (Yoon & Seger, 2006; Zeke et al, 2016; Unal et al, 2017).

The question of how MAP kinases (and other protein kinases) recognize their substrates is of considerable interest (Bardwell, 2006; Miller & Turk, 2018). The target site that MAPKs phosphorylate—a serine or threonine followed by a proline—is too degenerate to provide the requisite specificity for target recognition. Instead, MAPKs frequently tether themselves to their substrates by binding to a short linear motif designated a MAPK-docking site or "D-site" (Bardwell et al, 2009; Bardwell & Bardwell, 2015). D-sites in substrates are typically located near a cluster of target phosphosites, and D-site binding to MAPKs directs phosphorylation to these sites. The presence of a D-site, particularly if near a cluster of Ser–Pro or Thr–Pro (SP or TP) sites, can be used to identify putative novel substrates (Gordon et al, 2013; Zeke et al, 2015). Previously, during the course of such a bioinformatic search for novel human proteins containing predicted D-sites, we identified the GLI transcription factors as putative MAPK substrates (Whisenant et al, 2010).

Here, we demonstrate that MAPK-mediated phosphorylation of an N-terminal region of GLI1 near the major SUFU-binding site results in a dramatic weakening of SUFU binding to this region and a consequent release of repression. We identify three target phosphosites that are collectively necessary and sufficient for this effect, and we show that these three phosphorylated residues act cooperatively in SUFU release. In addition, we show that full-length GLI1 alleles containing phosphomimetic substitutions of these three residues are hyperactive in driving transcription in cells. Furthermore, we characterize mutations found in tumor samples that define an expanded SUFU-affinity-modulating region. Our findings suggest a mechanism that contributes to MAPK-mediated cross-activation of GLI-driven transcription.

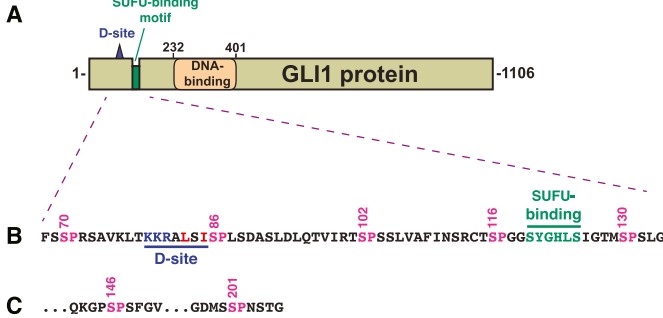

**Figure 1. Structure of GLI1 protein.**
**(A)** Schematic of the primary structure of human GLI1 protein, showing the MAPK-docking site (D-site), the SUFU-binding motif, and the zinc finger DNA-binding domain. The transcriptional activation domain constitutes a large fraction of the C-terminal half of the protein. **(B)** Amino acid sequence of residues 68–134, showing the D-site, the SUFU-binding motif, and a cluster of five canonical MAPK phosphorylation sites (SP). The numbers above show the position of the serine residues in the SP sites. **(C)** The last two of the seven SP MAPK phosphorylation sites contained in residues of 68–232 GLI1. After S201, the next SP or TP site occurs at S441.

# Results

## ERK2 phosphorylation of GLI1 modulates GLI1–SUFU binding

In a previous study, we exploited the observation that MAPKs briefly attach to many of their substrates before phosphorylating them to develop D-finder, a computational tool that searches genome databases for MAPK-docking sites (Whisenant et al, 2010). Among the novel substrates predicted by D-finder were human GLI1, GLI2 and GLI3. Using in vitro binding and protein kinase assays, we demonstrated that both ERK- and JNK-family MAPKs could bind to human GLI1 and GLI3 via the predicted D-site and phosphorylate nearby target residues. We also used mass spectrometry to determine that Ser343 in GLI3 was phosphorylated by ERK and JNK, and showed that the efficiency of this phosphorylation was dependent on the integrity of the D-site. Finally, we showed by mutagenesis that Ser130 in GLI1 (the comparable residue to Ser343 in GLI3) was also phosphorylated by both ERK and JNK (Whisenant et al, 2010).

As shown in Fig 1, the D-site and S130 phosphosite of GLI1 are located near the binding site for SUFU, a key negative regulator of GLI1 function (Hui & Angers, 2011). This arrangement of functional motifs suggested that MAPK-mediated phosphorylation might regulate GLI-SUFU binding. To investigate this hypothesis, we developed an in vitro GLI1–SUFU binding assay. To accomplish this, we first purified GST-GLI1$_{68-232}$, a fusion of residues 68–232 of the 1,106 residue human GLI1 protein to the GST affinity purification tag (Fig 2A). GLI1$_{68-232}$ contains the D-site, the SUFU binding site, and all nearby putative MAPK phosphosites, and is readily purified from bacteria. Following purification, varying amounts of GST-GLI1$_{68-232}$ protein were mixed with limiting amounts of full-length human SUFU protein that had been produced in radiolabeled form by in vitro transcription and translation. After a 15-min binding incubation, glutathione-Sepharose beads were added, and protein complexes were isolated by co-sedimentation and quantified by SDS–PAGE followed by Phosphorimager analysis. In this assay, SUFU

▶▶▶ Life Science Alliance

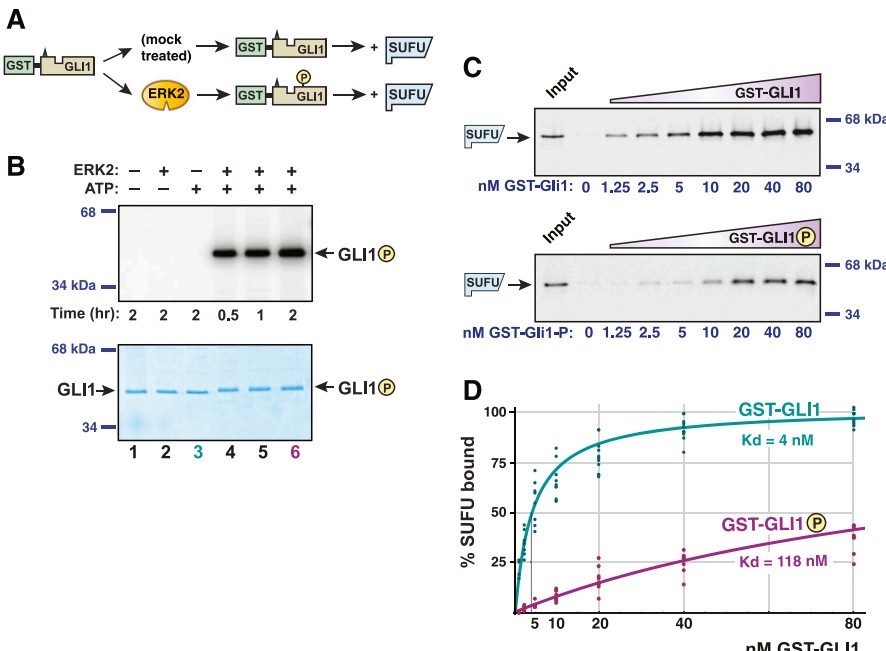

**Figure 2. GLI1–SUFU binding is weakened by ERK2-mediated phosphorylation of GLI1.**
**(A)** GST-GLI1, either treated with active ERK2 or mock treated, was tested for binding to radiolabeled SUFU protein. **(B)** Gel analysis GST-GLI1 phosphorylation. An aliquot of the samples used for the binding assays shown in Fig 2C and D was removed from the main reaction and spiked with $\gamma$-$^{32}$P-ATP to monitor the extent of ERK2-mediated phosphorylation; lane three corresponds to the mock treatment; lane six corresponds to the +ERK2 treatment; additional controls are also shown. The top panel is the autoradiographic image, the bottom panel is the same gel stained with Coomassie blue. GST-GLI1$_{68-232}$ migrated at its predicted mass of 44 kD; the position of the two closest molecular weight standards is shown on the right. A slight mobility shift is evident in the phosphorylated samples. **(C)** Gel analysis of SUFU binding to GST-GLI1. Approximately 1 pmol of $^{35}$S-radiolabeled full-length human SUFU protein was added to each binding reaction; 10% of this amount was loaded in the "Input" lane. The concentration of GST-GLI1 (mock or ERK2 phosphorylated) was varied from 0 to 80 nM (shown on bottom of gels; the GST-GLI1 used here was not radioactively labeled). After a 15-min incubation, bead-bound protein complexes were isolated by sedimentation and resolved by 10% SDS–PAGE. SUFU migrated at its predicted mass of 54 kD; the position of the two closest molecular weight standards is shown on the right. Gels were also stained for total protein using Coomassie blue to confirm equal loading/ sedimentation of mock versus ERK2-treated GST-GLI1; representative examples are shown in Fig S1. SUFU did not exhibit detectable binding to GST alone, even at a GST concentration of >1.7 $\mu$M (Fig S1). **(D)** GLI1–SUFU binding isotherms. Quantification of eight independent experiments of the type described in Fig 2C. Normalization and curve-fitting are as described in the Materials and Methods section. The scatter of the individual data points is also shown. Source data are available for this figure.

protein bound quite tightly to GLI1, with a Kd of ~4 nM (Fig 2). These results are consistent with other published measurements of the binding affinity between GLI1 and SUFU (Cherry et al, 2013; Szczepny et al, 2014).

Having identified the appropriate range of protein concentrations at which to test the GLI1–SUFU interaction, we developed a protocol to ask if MAPK-phosphorylated GLI1 exhibited diminished binding to SUFU when compared with unphosphorylated GLI1 (Fig 2A). Purified GST-GLI1$_{68-232}$ was first phosphorylated to high stoichiometry (~3 mol phosphate per mole substrate) with purified active ERK2. During this process, part of each sample was removed and spiked with radioactive ATP to quantify phosphate incorporation (Fig 2B). The kinase reaction was then halted with EDTA, and glutathione-Sepharose beads were added to capture the GLI1. After the removal of the kinase and ATP by extensive washing of the beads, radiolabeled SUFU was added and a binding assay was performed. A parallel "mock" sample was treated identically except that the ERK2 enzyme was omitted. As shown in Figs 2C and S1, unphosphorylated GLI1 bound with high affinity to SUFU, whereas ERK2-phosphorylated GLI1 displayed substantially reduced binding affinity. Quantification of eight independent experiments revealed that the GLI1–SUFU dissociation constant was increased almost 30-fold by ERK2-mediated phosphorylation of GLI1 (Fig 2D). Note that increases in the dissociation constant (Kd) correspond to proportional decreases in affinity, as Kd is the reciprocal of the affinity constant. Similarly, increases in Kd correspond to decreases in the amount of protein complex formed. For example, a 10-fold increase in Kd can result in up to a 90% decrease in amount of

heterodimer formed, and a 30-fold increase in Kd can result in up to a 97% decrease, depending on the concentrations of the interacting proteins relative to the Kd of their interaction. For convenience, we hereafter refer to the ability of ERK2-mediated phosphorylation of GLI1 to weaken the GLI1–SUFU interaction as "ERK2-mediated SUFU release," or more simply, "SUFU release."

## Phosphorylation occurs on canonical sites and is necessary for SUFU release

MAPKs are proline-directed kinases, and thus, in the vast majority of cases, phosphorylate their substrates on serines or threonines that are immediately followed by prolines (i.e., SP or TP). GST-GLI1$_{68-232}$ contains seven SP motifs (S70, S86, S102, S116, S130, S146, and S201), and no TP motifs. To determine if phosphorylation of one or more of these seven sites was necessary for ERK2-mediated SUFU release, we constructed a mutant version of GST-GLI1$_{68-232}$, designated GLI1$^{7A}$, in which all seven serines in these SP motifs were changed to alanines by site-directed mutagenesis.

After purification, both wild-type GLI1 and GLI1$^{7A}$ were then either treated with purified active ERK2 or mock treated, and then assessed for binding to SUFU (Fig 3A). During this process, part of each sample was removed and spiked with radioactive ATP to quantify phosphate incorporation. As shown in Fig 3B, compared with wild-type GLI1$_{68-232}$, GLI1$^{7A}$ was only minimally phosphorylated by ERK2, indicating that the vast majority of ERK-mediated phosphorylation occurs at one of more of the seven canonical SP sites.

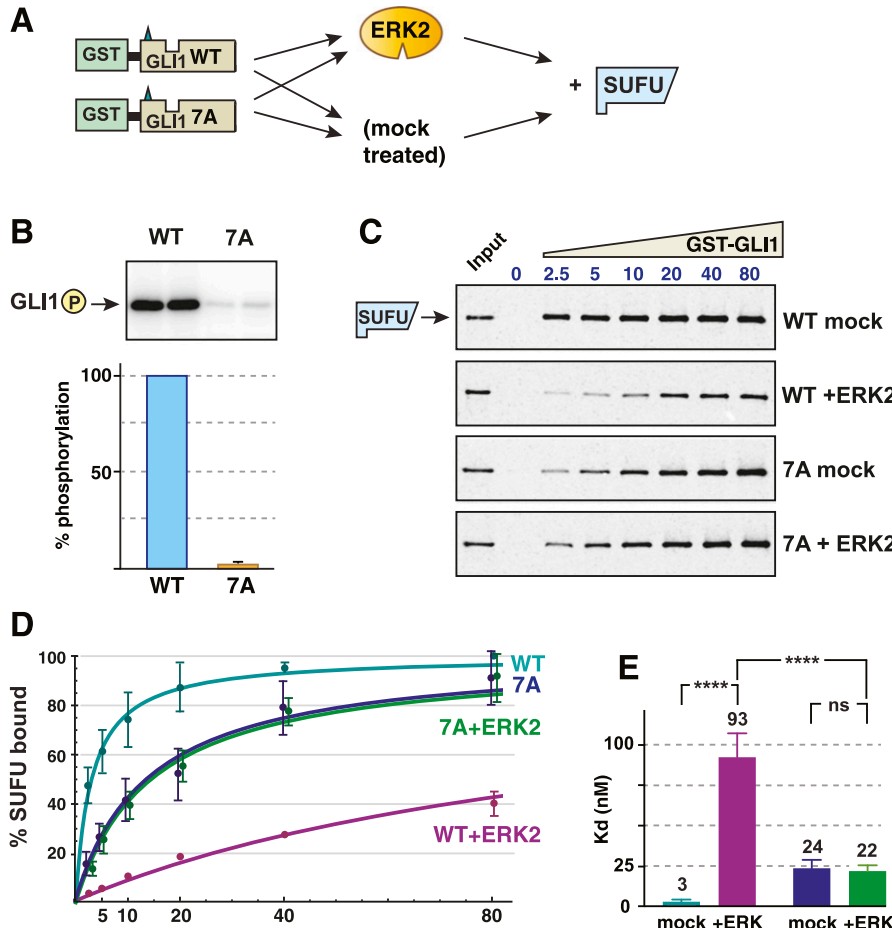

**Figure 3. Phosphorylation of SP residues is required for SUFU release.**
**(A)** Wild type GST-GLI1$_{68-232}$ and GST-GLI1$^{7A}$ were either treated with active ERK2 or mock treated, and then tested for binding to radiolabeled SUFU protein. **(B)** ERK2 kinase assay using wild-type GST-GLI1 (WT) or GST-GLI1$^{7A}$ (7A) as the substrate. Top: Analysis by SDS–PAGE and autoradiography; replicate samples are shown. Bottom: quantification, n = 5, with wild-type phosphorylation normalized to 100%, the error bar shows the 95% confidence interval. **(C)** GLI1–SUFU binding assay, representative experiment, analyzed by SDS–PAGE and autoradiography. Other details as in Fig 2C. Gels were also stained for total protein using Coomassie blue to confirm equal loading/sedimentation of mock versus ERK2-treated GST-GLI1 (Fig S2). **(D)** GLI1–SUFU binding isotherms. The data points and the fitted curves for 7A (mock-treated) and 7A+ERK2 are offset slightly for visual clarity. Error bars show the 95% confidence interval; bars that were less than 20% of the mean are not shown. **(E)** Comparison of Kd values for the four conditions. Each data point in the five experiments was converted to a Kd value as described in the Materials and Methods section, n = 30. Numbers above each bar are the mean Kd in nanomolar units; error bars show the 95% confidence interval. ****$P < 0.0001$; ns, not significant ($P > 0.05$). Higher Kd's equate to lower levels of binding.

When assessed for SUFU binding, there was a substantial difference between phosphorylated versus mock-treated wild-type GLI1 (Figs 3C and S2), consistent with the findings presented in Fig 2. In contrast, there was no observable difference in the affinity of ERK2-treated versus mock-treated GLI1$^{7A}$ for SUFU (Fig 3C). When the results of five independent experiments were quantified, it was apparent that mock-treated GLI1$^{7A}$ displayed a somewhat reduced affinity for SUFU compared with mock-treated wild-type GLI1 (Fig 3D). This suggested that one or more of the seven Ser-to-Ala mutations in GLI1$^{7A}$ marginally destabilizes the GLI1–SUFU interaction, perhaps by slightly altering the conformation of the binding domain. Indeed, further experiments, reported below, indicated that the S102A substitution was the major contributor to this destabilization. Quantification of the effect on the dissociation constant is shown in Fig 3E. In this set of experiments, SUFU bound to unphosphorylated wild-type GST-GLI1 with Kd of 3 nM, and this was weakened by over 30-fold when GLI1 was phosphorylated by ERK2. In contrast, both mock-treated and ERK2-treated GLI1$^{7A}$ bound to SUFU with a Kd of ~23 nM. Importantly, treatment of GLI1$^{7A}$ with active ERK2 had no effect on SUFU binding. Furthermore, whereas the difference in the Kd for SUFU binding between mock-treated and ERK2-treated wild-type GLI1 was highly significant ($P < 0.0001$), the difference between mock-treated and ERK2-treated GLI1$^{7A}$ was

not statistically significant ($P = 0.57$). Based on these results, we concluded that ERK2-mediated phosphorylation at one or more of the seven SP sites present in GLI1$_{68-232}$ is required for ERK2-mediated SUFU release.

## Ser102, Ser116, and Ser130 are efficiently phosphorylated by ERK2

To measure the phosphorylation efficiency of each of the seven SP sites individually, a series of "single phosphosite only" mutants were constructed. In these seven mutants, only one of the seven SP phosphorylation sites remains intact, and the other six are mutated to alanine. Each of these mutant derivatives was then purified individually and tested in kinase assays. As determined by the incorporation of the γ phosphate of radioactive ATP in the presence of active ERK, the most efficient target sites in GLI1 were Ser116 and Ser130, followed by Ser102 and Ser201. Phosphorylation of the other three sites was minimal (Fig 4).

## Phosphorylation of S102, S116, and/or S130 is necessary and sufficient for SUFU release

We focused our further analysis on S102, S116, and S130, as these three residues are the closest to the SUFU-binding motif (Fig 1), and

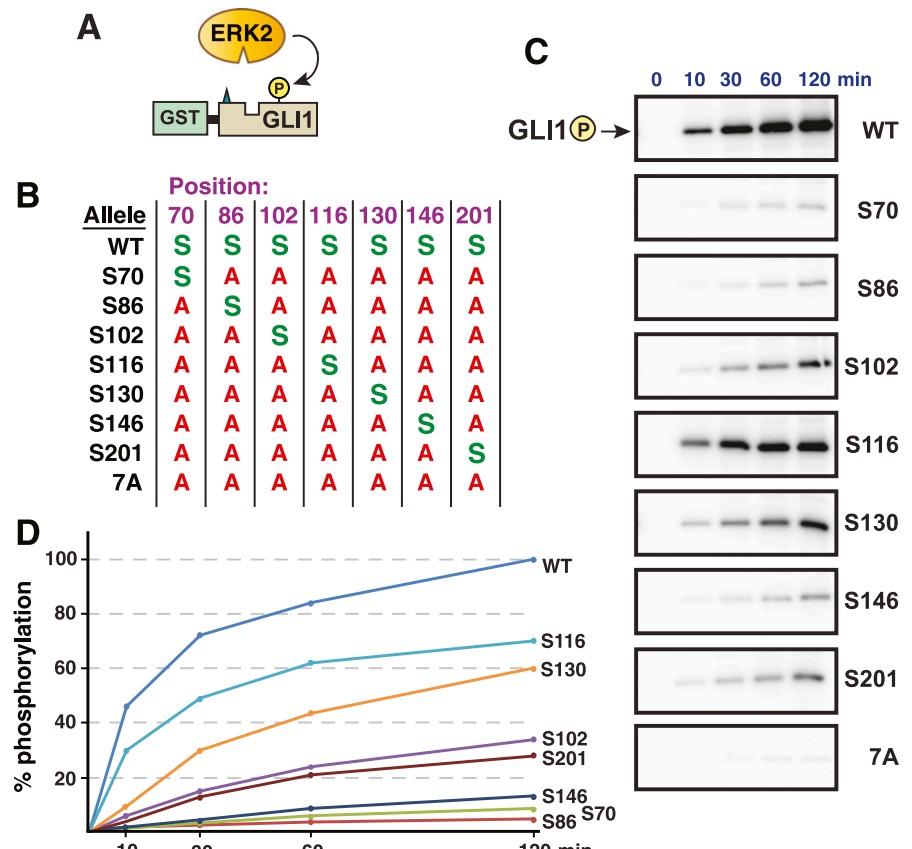

**Figure 4. ERK2-mediated phosphate incorporation at individual SP sites in GLI1.**
**(A)** Purified GST-GLI1$_{68-232}$ wild-type or mutant proteins were mixed with purified active ERK2 and radiolabeled ATP, and assessed for phosphorylation efficiency. **(B)** Chart detailing the substitutions present in site-directed mutants of GST-GLI1$_{68-232}$. The position of the residue is shown on the top. An *S* indicates that the native serine residue is present at that position; an *A* indicates an alanine substitution at that position. **(C)** Gel analysis of time course of phosphate incorporation in a kinase assay using 10 units of ERK2. Incubation time is shown on the top; allele designation is on the right. All variants migrated at the expected molecular weight for GST-GLI1$_{68-232}$ (44 kD). **(D)** Time course of GLI1 phosphorylation, average of three experiments such as shown in Fig 4C, normalized by setting phosphorylation of the wild-type allele at 120 min as 100%. Error bars (not shown for visual clarity) were less than 20% of the corresponding means.

were also the three sites most efficiently phosphorylated by ERK2 (Fig 4). To ask if the integrity of these three sites was necessary for ERK-mediated SUFU release, we constructed a mutant derivative in which all three of these serines (and only these serines) were substituted with alanine (GLI1$^{AAA}$, Fig 5A). This mutant, when mock-treated, bound to SUFU with a Kd of 17 nM (Figs 5B and C and 6A). Notably, upon phosphorylation, the binding of this mutant did not appreciably change (Kd 20 nM); that is, there was no detectable release of SUFU. This result demonstrates that phosphorylation of one or more of the residues S102, S116, and/or S130 is necessary for SUFU release. In addition, because S70, S86, S146, and S201 remained intact in this mutant, this result also demonstrates that phosphorylation of these four residues plays little or no role in promoting SUFU release.

To ask if the presence of these three MAPK phosphosites was sufficient for ERK-mediated SUFU release, we constructed a mutant derivative of GLI1$_{68-232}$ in which only S102, S116 and S130 were left intact, and the other MAPK phosphosites (S70, S86, S146, and S201) were mutated to alanine (Fig 5A). This mutant, which we designated GLI1$^{SSS}$, was then either phosphorylated by active ERK2 or mock treated (as shown in Fig 3A for WT versus 7A), and then tested for binding to radiolabeled SUFU. As shown in Fig 5C, mock-treated GLI1$^{SSS}$ protein bound to SUFU with a Kd of 2.6 nM (rounded to 3 nM in Fig 5C), essentially indistinguishable from the Kd of 4 nM observed for wild-type GLI1 tested in parallel in the

same set of experiments. Likewise, ERK-phosphorylated GLI1$^{SSS}$ protein was efficiently released from SUFU, as shown by a Kd increase from 3 to 127 nM upon phosphorylation; this magnitude was even greater than the increase to 93 nM observed for wild-type GLI1. Thus, the binding of both wild-type GLI1 and GLI1$^{SSS}$ to SUFU was weakened by more than 35-fold by ERK2-mediated phosphorylation, resulting in efficient SUFU release (Figs 5B and C and 6B). These results indicate that phosphorylation on one or more of the residues S102, S116 and/or S130 is sufficient for SUFU release. In addition, because serines 70, 86, 146, and 201 were mutated to alanine in this mutant, this result also demonstrates that phosphorylation of one or more of these four residues is not required for SUFU release. Statistical analysis of these results is presented further below.

Although the above experiments seemed to rule out a major role for S201 in SUFU release, we noted that the efficiency of phosphorylation of S201 seen in Fig 4 was almost indistinguishable from S102. Thus, we wondered if phosphorylation of S201 might potentiate the level of release seen with the GLI1$^{SSS}$ variant. To address this possibility, we constructed an additional variant designated GLI1$^{SSSS}$. In GLI1$^{SSSS}$, S102, S116, S130, and S201 are intact, whereas S70, S86, and S146 are mutated to alanine. As shown in Fig S4, GLI1$^{SSSS}$ did not exhibit a greater level of SUFU release than GLI1$^{SSS}$. This result confirms that phosphorylation of S201 does not play a major role in SUFU release.

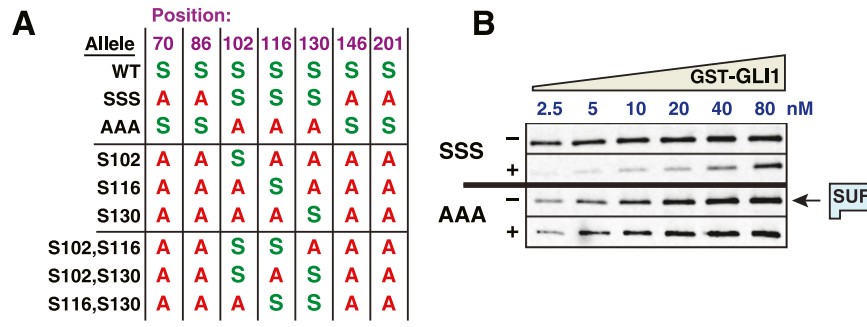

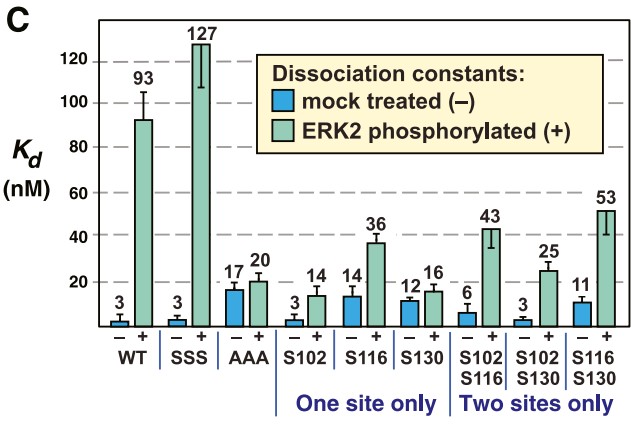

**Figure 5. Multisite phosphorylation triggers SUFU release.**
**(A)** Chart detailing the substitutions introduced into GST-GLI1$_{68-232}$. An $S$ indicates that the native serine residue is present at the position shown on the top in the allele shown on the left; an $A$ indicates an alanine substitution at that position. **(B)** GLI1–SUFU binding assay. Purified GST-GLI1$^{SSS}$ and GST-GLI1$^{AAA}$ were mock treated (–) or phosphorylated by ERK2 (+), mixed with limiting amounts of radiolabeled SUFU protein, and SUFU binding was analyzed by SDS–PAGE and autoradiography. Autoradiographs of additional variants are shown in Fig S3. Gels were also stained for total protein using Coomassie blue to confirm equal loading/sedimentation of mock versus ERK2-treated GST-GLI1 variants; representative examples are shown in Fig S3. **(C)** Quantification of dissociation constants (Kd). Data are from 3 to 5 independent experiments of the type shown in Fig 5B. Each data point in the five experiments was converted to a Kd as described in the Materials and Methods section, n = 18–30. Numbers above each bar are the mean Kd in nanomolar units; error bars show the 95% confidence interval. See text for statistical analysis. Higher Kd's equate to lower binding affinity.
Source data are available for this figure.

## Phosphorylated residues cooperate to promote SUFU release

To delineate the differential contribution of phosphorylated S102, S116, and S130 to SUFU release, we constructed a series of six additional mutant versions of GST-GLI1$_{68-232}$ containing serine to alanine substitutions of one or two of these three critical residues. In addition, each of these mutants also had serine to alanine substitutions at residues 70, 86, 146, and 201, so that in total they contained only a single MAPK-phosphorylatable serine (as in Fig 4) or two phosphorylatable serines (Fig 5A).

The results of the investigation of the full set of nine alleles (eight mutant alleles plus wild type) is shown in Figs 5C, 6A–H, and S3. This analysis revealed that all four alleles carrying an S102A substitution (GLI1$^{AAA}$, GLI1$^{S116}$, GLI1$^{S130}$, and GLI1$^{S116,S130}$; see Fig 5A) displayed weaker binding to SUFU than the five alleles in which the native serine at position 102 was maintained. These findings are consistent with the results obtained with the 7A allele (see Fig 3), and suggest that replacing GLI1 Ser102 with alanine weakens binding to SUFU, increasing the dissociation constant by about 3.5-fold.

Despite this complication, the outcome of this set of experiments was quite clear. First, the results obtained with GLI1$^{S102}$, GLI1$^{S116}$, and GLI1$^{S130}$, the three mutants where "one site only" remained intact, indicated that the phosphorylation of any single residue was not sufficient for full SUFU release. For example, mock-treated GLI1$^{S102}$ bound to SUFU with an affinity comparable with mock-treated wild-type GLI1 and mock-treated GLI1$^{SSS}$ (Kd ~3 nM; Fig 5C). Upon phosphorylation, however, the Kd of GLI1$^{S102}$ increased by less than fivefold, to 14 nM. Similarly, mock-treated GLI1$^{S116}$ bound to SUFU with a Kd of 14 nM, and this increased by 2.6-fold upon

phosphorylation. For GLI1$^{S130}$, there was only a 1.3-fold reduction in SUFU-binding affinity after ERK2 treatment. These modest changes in binding affinity are substantially less than the 35-fold change observed after ERK2 phosphorylation of GLI1$^{SSS}$, in which all three key phosphorylation sites are intact.

Three mutant constructs contained two of the three residues S102, S116 or S130 intact, with the third mutated to alanine (these are labeled "two sites only" in Fig 5C). These mutants, when phosphorylated by ERK2, exhibited levels of SUFU release that was greater than the corresponding "one site only" mutants, but substantially less than wild-type or GLI1$^{SSS}$. For the three "one site only"-intact mutants, the Kd increased an average of threefold after ERK2-mediated phosphorylation (range of 1.3–4.7). For the three "two sites only"-intact mutants, the Kd increased about sevenfold after ERK2-mediated phosphorylation (range 4.8–8.3). Only when all three sites were intact (wild-type and GLI1$^{SSS}$) was the dramatic 35+-fold increase in Kd observed upon phosphorylation. Statistical analysis of these results is presented in the next section.

Taken together, these results indicate that phosphorylation of all three residues, S102, S116, and S130, are required for the level of SUFU release exhibited by wild-type GLI1. Phosphorylation of any single residue is not sufficient for full SUFU release, nor is phosphorylation of any two of the three residues.

## Statistical analysis of the effect of multisite phosphorylation on SUFU release

To assess the statistical significance of the results shown in Figs 5 and 6, we used Welch's $t$ test, which does not assume that the

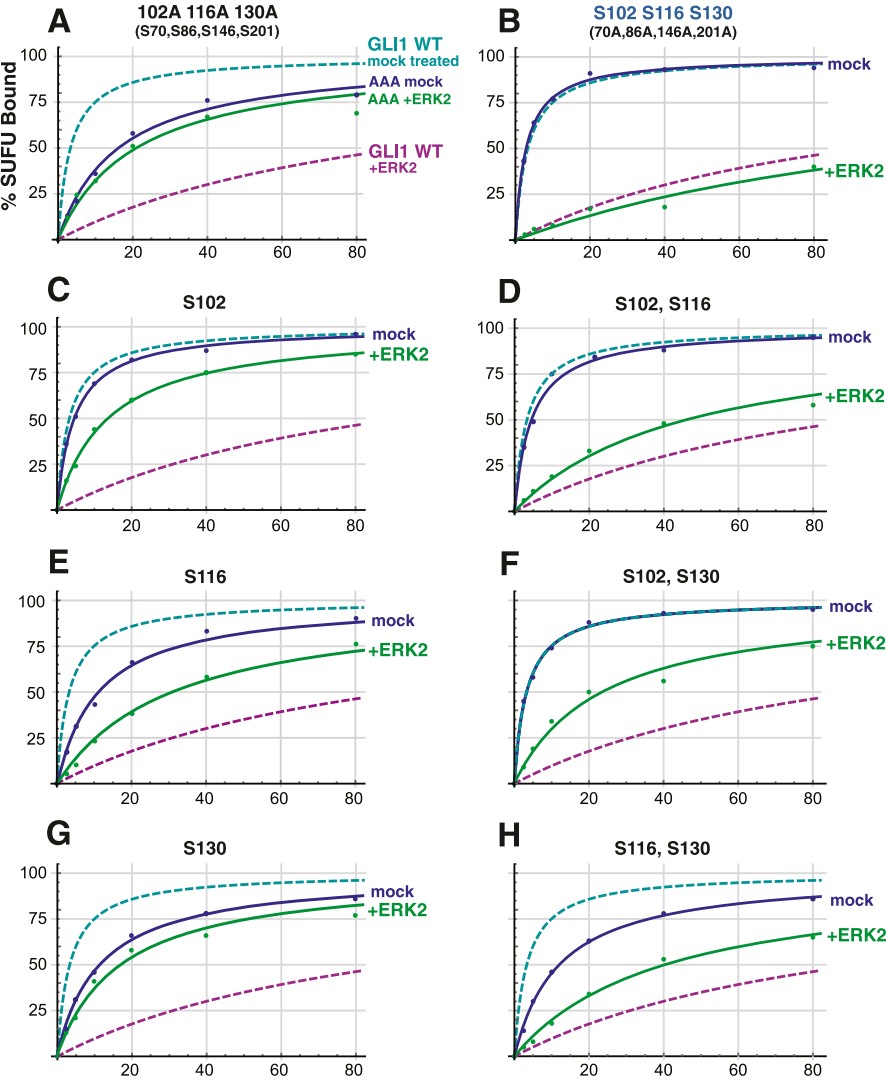

**Figure 6. Phosphorylated residues cooperate to promote SUFU release.**
**(A)** Binding isotherms of mock-treated and ERK2-treated GST-GLI1[AAA]. Data are from 3 to 5 independent experiments of the type shown in Fig 5B. The isotherms for mock-treated and ERK2-phosphorylated wild-type GST-GLI1 are shown for comparison. **(B)** Binding isotherms of mock-treated and ERK2-phosphorylated GST-GLI1[SSS]. Data are from 3 to 5 independent experiments of the type shown in Fig 5B. The isotherms for mock-treated and ERK2-phosphorylated wild-type GST-GLI1 are again shown for comparison. **(C, D, E, F, G, H)** Binding isotherms of mock-treated and ERK2-phosphorylated GST-GLI1 single- and double-phosphosite variants. Data are from 3 to 5 independent experiments of the type shown in Fig 5B. The isotherms for mock-treated and ERK2-phosphorylated wild-type GST-GLI1 are shown for comparison. **(C)** S102 (only S102 is intact, the other SP phosphosites are mutated to alanine). **(D)** S102, S116. **(E)** S116. **(F)** S102, S130. **(G)** S130. **(H)** S116, S130.

different populations sampled have the same variance (Ruxton, 2006). Welch's test is more conservative, that is, less likely to lead to false positives, than the standard $t$ test. Given that we planned to perform multiple comparisons, it was also important to apply a correction for multiple hypothesis testing. We chose to use the Bonferroni correction, which is the most conservative correction (Abdi, 2007). Applying this correction, an adjusted $P$-value threshold for 95% confidence of 0.0025 (=0.05/20) was chosen because we planned a total of 20 comparisons. The Bonferroni correction makes the threshold for significance more stringent by a factor equal to the number of planned comparisons; see the Materials and Methods section for further details.

First, we determined for each mutant allele, if the observed increase in the Kd for SUFU-binding upon phosphorylation was significant (i.e., we compared each mutant to itself, mock-treated versus phosphorylated, for a total of eight comparisons). For all mutants except AAA ($P$ = 0.18) and S130-only ($P$ = 0.006, which does not fall below the Bonferroni-corrected threshold of 0.0025), the observed Kd increase upon phosphorylation was highly significant ($P$ < 0.0001).

Next, we asked if the Kd of 130 ± 20 nM that we measured for the phosphorylated GLI1[SSS] mutant, which has all three key phosphosites intact, was significantly different from the Kds of the phosphorylated single and double-phosphosite mutants (which ranged from 14 ± 4 to 53 ± 10 nM; total of six comparisons). Indeed, the difference between the 130 nM Kd and any of the other six Kds was highly significant ($P$ < 0.0001). Thus, when all three key phosphosites are intact, there is significantly more SUFU release than when any two of the three are present, or any single one.

The final set of six comparisons we performed was to compare each phosphorylated "one site only" mutant to each of the "two sites only" mutants of which it was part. For example, we compared the Kd of ERK2-treated GLI1[S102] (14 ± 4 nM) to the Kd of ERK2-treated GLI1[S102,S116] (43 ± 8 nM), as well as to the Kd of ERK2-treated GLI1[S102,S130] (25 ± 3 nM). This analysis revealed that GLI1[S102] was significantly different than both of the corresponding "two sites only" mutants, as was GLI1[S130] (all $P$-values < 0.001). However, GLI1[S116] was not significantly different from either GLI1[S102,S116]

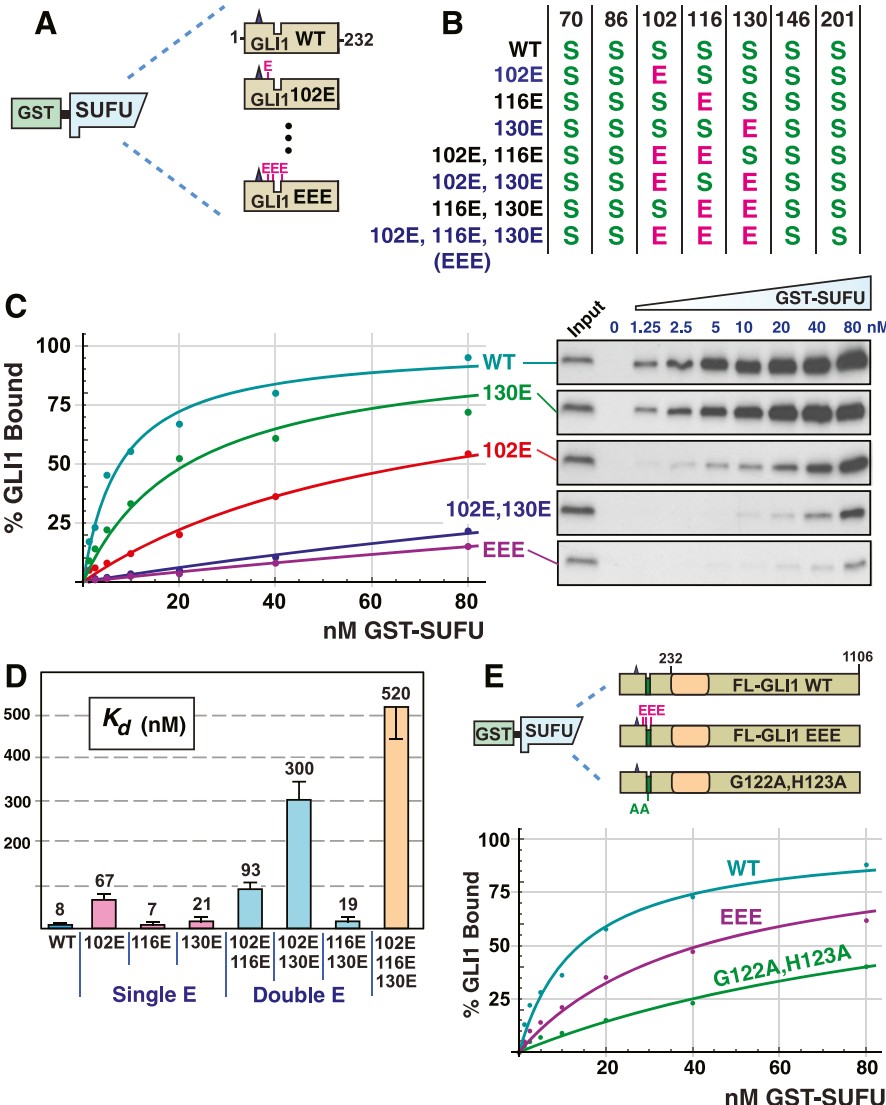

**Figure 7. Phosphomimetic substitutions decrease GLI1 binding to SUFU.**
**(A)** Radiolabeled GLI1$_{1-232}$ variants were assessed for binding to GST-SUFU protein. Radiolabelled GLI1 does not bind appreciably to GST alone, even when 40 μg of GST is used (Whisenant et al, 2010). **(B)** Chart detailing the phosphomimetic substitutions present in site-directed mutants of GLI1$_{1-232}$. S indicates that the native serine residue is present at the position shown on the top; E indicates a glutamic acid substitution at that position. **(C)** GLI1–SUFU binding isotherms (left) and gel analysis of a representative experiment (right). The data points on the graph are an average of four experiments. Only data for selected alleles is shown. Gels were also stained for total protein using Coomassie blue to confirm equal loading/sedimentation of GST-SUFU; representative examples are shown in Fig S5. **(D)** Quantification of dissociation constants (Kd) for all eight GLI1$_{1-232}$ alleles. Data are from four independent experiments. Each data point in the four experiments was converted to a Kd as described in the Materials and Methods section, n = 28. Numbers above each bar are the mean Kd in nanomolar units; error bars show the 95% confidence interval. See text for statistical analysis. **(E)** Radiolabeled, full-length GLI1 variants were assessed for binding to GST-SUFU protein. The data points on the graph are an average of four experiments.

(*P* = 0.09) or GLI1$^{S116,S130}$ (*P* = 0.003, which barely misses the Bonferroni-corrected threshold of 0.0025).

### Phosphomimetic substitution of GLI1S102, S116, and S130 decrease GLI1 binding to SUFU

To assess the involvement of GLI1 residues S102, S116, and S130 in SUFU binding using a different approach than shown in Figs 2–6, we substituted one, both, or all three of these key serine residues with the amino acid glutamate, which resembles phosphoserine in its size and negative charge. To keep our analysis focused on the N-terminus of GLI1, these substitutions were made in a plasmid encoding GLI1 residues 1–232 downstream of a T7 RNA polymerase promoter (Fig 7A and B).

Next, full-length human SUFU protein was produced in, and purified from, *Escherichia coli* bacteria as an N-terminal GST fusion protein. The purified GST-SUFU was then tested for binding to wild-type GLI1$_{1-232}$ that had been produced and radiolabeled by in vitro transcription/translation, as well as to the various GLI1$_{1-232}$ phosphomimetic substitution alleles (Fig 7A).

As shown in Figs 7C and D and S5, GST-SUFU bound tightly to radiolabeled wild-type GLI1$_{1-232}$ (Kd 8 nm ± 2). Indeed, the affinity of this interaction was comparable with the previous experiments (Figs 2, 3, 5, and 6) assessing GST-GLI1$_{68-232}$ binding to radiolabeled full-length SUFU. Of the three single phosphomimetic substitutions, S102E had the largest effect, reducing GLI1–SUFU binding affinity by about eightfold. The S102E S130E double mutant had a larger effect (38-fold). Furthermore, the S102E S116E S130E triple mutant had the largest effect (65-fold). Indeed, with one exception, every double mutant had a greater effect than either of the corresponding single mutants (the exception is that S116E, S130E was not different from S130E alone, as detailed further below). Moreover, the triple mutant had a greater effect than any of the three double mutants. Thus, phosphomimetic substitutions in these three residues cooperated to weaken SUFU binding, and this cooperation was super-additive, or synergistic, with regard to the effect on binding affinity.

To determine statistical significance, we performed the following planned comparisons: (1) the wild type was compared with each of seven mutants; (2) each of the three double mutants was compared with the two corresponding single mutants (for instance, the 102E,116E double mutant was compared with the 102E single mutant and to the 116E single mutant); and (3) the EEE triple mutant was compared with each of the three double mutants. These 16 comparisons were performed using Welch's $t$ test, as was done for the analysis in Figs 5 and 6. Given that multiple comparisons were being performed, it was also important to apply a correction for multiple hypothesis testing. Again, we chose to use the Bonferroni correction, which is the most conservative correction. Applying this correction, a $P$-value threshold of 0.003 (=0.05/16) was chosen. All but two of the 16 comparisons were significant at this level: S116E was not significantly different than wild type ($P = 0.66$), and S116E, S130E was not significantly different than S130E ($P = 0.47$). Although phosphomimetic substitution of S116 had no effect on its own, or in combination of with S130E, it did have an effect in two other contexts. First, S102E, by itself, reduced GLI1–SUFU binding affinity by about eightfold, whereas the S102E S116E double mutant had a slightly larger effect (12-fold). Second, as mentioned above, the S102E S116E S130E triple mutant had a much larger effect on SUFU binding (65-fold increase in Kd) than the S102E S130E double mutant (38-fold).

## Phosphomimetic substitutions have a reduced effect on full-length GLI1

The experiments presented in Figs 2 and 4–6 show that ERK2-mediated phosphorylation of key residues near the MAPK-docking site and SUFU-binding motif have a dramatic effect on the ability of GLI1$_{68-232}$ to bind to full-length SUFU. Similarly, the experiments shown in Fig 7 demonstrate that phosphomimetic substitution of those same key residues in the context of GLI1$_{1-232}$ have a dramatic effect on the ability of GLI1$_{1-232}$ to bind to full-length GST-SUFU. We were unable to purify full-length, soluble, GLI1 protein from bacteria, and thus we could not biochemically assess the effect of ERK2-mediated phosphorylation on the ability of full-length GLI1 to bind to SUFU. Nevertheless, our ability to purify full-length GST-SUFU from bacteria, and to produce full-length, soluble GLI1 (and mutants thereof) by in vitro transcription/translation, allowed us to assess the effect of phosphomimetic substitutions on the binding affinity of full-length GLI1 to full-length SUFU (Fig 7E).

As shown in Fig 7E, full-length GLI1 protein bound to GST-SUFU with an affinity (Kd = 14 nM ± 4) comparable with the affinity that GLI1$_{1-232}$ protein bound to GST-SUFU (Kd = 8 nM ± 2). In contrast, the reduction in affinity caused by the S102E S116E S130E triple mutation was less dramatic in the context of full-length GLI1 (Kd = 42 nM ± 8, a threefold decrease in affinity) than in the context of GLI1$_{1-232}$, where there was a 65-fold decrease in affinity. To provide a further point of comparison, we introduced the G122A, H123A double substitution into full-length GLI1. These substitutions lie in the core of the $^{120}$SYGHLS$^{125}$ SUFU-binding motif, and were previously shown to substantially reduce the binding of SUFU to the N-terminus of GLI1 (Dunaeva et al, 2003). Full-length GLI1$^{G122A,H123A}$ displayed about 10-fold reduced affinity for GST-SUFU (Kd = 120 nM ± 35).

## Some GLI1 somatic mutations found in tumors reduce binding to SUFU

Our finding that SUFU binding was diminished by phosphorylation of, or phosphomimetic substitution of, GLI1S102, S116, and S130, led us to ask if the region between the D-site and S130 might either (a) be a part of an extended N-terminal SUFU-binding region that was larger than the recognized minimal SUFU-binding motif (GLI1 residues 120–125), or (b) communicate allosterically with this motif. Consistent with these possibilities, the region between and including the D-site and the SUFU-binding motif is conserved between human GLI1, GLI2, and GLI3 (Fig 8C), and, as shown in the next section, is also conserved between GLI orthologs present in most of the major animal phyla (Fig 9).

To begin to investigate these possibilities, we chose 20 tumor-derived single-amino-acid substitution mutations in the region spanning residues 80–130 of GLI1, and used site-directed mutagenesis to introduce them into GLI1$_{1-232}$. These mutations were chosen from those documented in the COSMIC (catalog of somatic mutations in cancer) database (Tate et al, 2019), as well as from mutations found from targeted sequencing of the *GLI1* gene in advanced or drug resistant basal cell carcinomas (Table S1). Mutant derivatives were then produced by in vitro transcription/translation and tested for binding to GST-SUFU (Fig 8A). The relative binding of each mutant was compared with wild-type GLI1$_{1-232}$, which was used as a positive control in every experiment. Because this procedure resulted in 20 planned comparisons, we used a Bonferroni correction factor of 20, resulting in an adjusted cut off for 95% confidence of 0.05/20 = 0.0025.

In this set of experiments, wild-type GLI1$_{1-232}$ bound to GST-SUFU with a Kd of 14 ± 4 nM. 16 of the 20 tumor-derived point mutations had no discernible effect on the binding of GLI1$_{1-232}$ to full-length GST-SUFU. This set included mutations in and around the D-site, including R81W, S84P and L88M, and S89L. The R81Q mutation had a small but reproducible and statistically significant effect on binding (Kd = 37 ± 10, a ~2.5-fold decrease in affinity; $P < 0.001$; Fig 8B). These results suggest that the MAPK-docking site is not directly involved in SUFU binding, although a more comprehensive mutational scan will be required to definitively establish this. Likewise, tumor-derived substitutions in a stretch of 12 amino acids running from V107 to G118 did not affect binding to SUFU.

We tested two tumor-derived mutations in residues that flank S102: R100C and S105F (Fig 8B). R100C mutations have been found in a basal cell carcinoma, a head/neck squamous cell carcinoma and two different lung adenocarcinomas. This mutant exhibited a substantially reduced affinity for GST-SUFU (Kd = 498 ± 81, a ~35-fold decrease in affinity). S105 somatic mutations have been found in five tumors: S105F in a head/neck cancer and a B cell leukemia, S105T in a liver cancer, and S105Y in two different lung cancers. The S105F mutation had a major effect on binding SUFU (Kd = 181 ± 30 nM, a ~13-fold decrease in affinity). For the purposes of comparison, the S102A single mutation decreased affinity for SUFU by 3.5-fold (Fig 6), and the S102E single mutation decreased affinity for SUFU by eightfold (Fig 7).

One tumor-derived mutation, L124P, was within the $^{120}$SYGHLS$^{125}$ SUFU-binding motif. This mutation, from a drug-resistant basal cell carcinoma patient, dramatically diminished

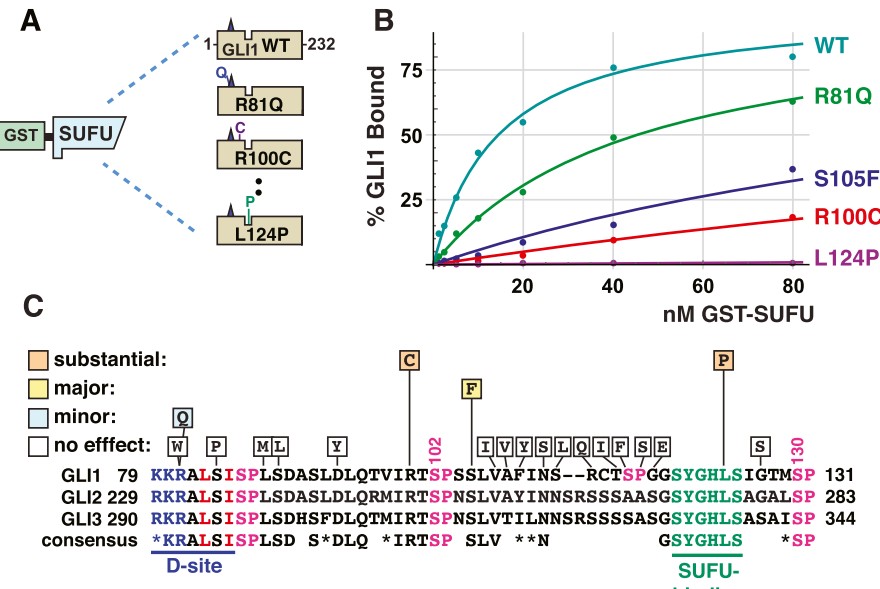

**Figure 8. Effect of tumor–derived mutations on GLI1 binding to SUFU.**
**(A)** Radiolabeled GLI1$_{1-232}$ variants were assessed for binding to GST-SUFU protein. **(B)** GLI1–SUFU binding isotherms of selected mutants. The data points on the graph are an average of three or four different experiments. **(C)** The chart shows which single substitutions in GLI1$_{1-232}$ had no effect (defined as an affinity change that was either less than twofold, and/or was not statistically significant), a minor effect (affinity decrease between two and fivefold that was statistically significant), a major effect (greater than 10-fold, significant), or a substantial effect (greater than 35-fold, significant). An alignment of human GLI1 residues 79–131 with the corresponding regions of human GLI2 and GLI3 is also shown, with the consensus shown below. Asterisks (*) indicate chemically similar residues that are also frequently substituted for one another in homologous sequences (Pearson, 1990).

GLI1–SUFU binding (Kd = 3,800 ± 1,600 nM, a ~250-fold decrease in affinity; Fig 8B).

Collectively, these results affirm the critical importance of the $^{120}$SYGHLS$^{125}$ SUFU-binding motif (Dunaeva et al, 2003), yet additionally suggest that a small region that includes S102 and nearby residues (R100, S105) is also involved in SUFU binding.

### The MAPK D-site and target phosphosites are evolutionarily conserved

We examined vertebrate orthologs of GLI1 for conservation of five motifs–the D-Site, the SP102, SP116, and SP130 phosphosites, and the SUFU-binding motif (Fig 9A). For convenience, we will hereafter use the term "SP102" to refer to Ser102-Pro103 in human GLI1, SP130 to refer to Ser130-Pro131 in human GLI1, etc. Moreover, we will also use the term "SP102" to refer to the equivalent position in all orthologs, even though the residue numbers will naturally be different in these orthologs. Examination of 149 mammalian species indicated that all placental mammals possess all five of these motifs, with zero or minimal sequence changes when compared with human GLI1. The four marsupial species we examined, however, have PP rather than SP116. We also compared 61 bird species, 18 reptilian species, 6 amphibian species, and 74 fish species. We found that birds, reptiles, amphibians and fish possess four of the five motifs, with close agreement to the mammalian consensus, but—like marsupials—do not possess a MAPK phosphorylation site equivalent to SP116. The terrestrial classes lack the proline in the SP116 site, whereas fish lack the serine as well. Hence, SP116 may be an invention of placental mammals, whereas the other four motifs are widely conserved in vertebrates.

We looked for conservation of the five motifs in the two non-vertebrate chordate subphyla—tunicates (e.g., sea squirts) and cephalochordates (lancelets)—as well as in the eight major invertebrate phyla (Fig 9B). In most non-vertebrate animal species,

there is only one *GLI* gene, as the three vertebrate paralogs arose from two gene duplication events early in vertebrate evolution (Abbasi et al, 2009). Hence, we were not surprised to find that the SP116 site is not conserved in invertebrates, as it appears to have originated in placental mammals. In contrast, a very close or exact fit to the D-site is present in chordates, echinoderms (e.g., sea urchins) and one of the most ancient animal phyla, Porifera (sponges). These D-sites are predicted to be fully functional for MAPK docking, as they contain an intact basic submotif (R/K-R-K in vertebrate GLI proteins) as well as an intact hydrophobic submotif (LSI in vertebrate GLI proteins). Both of these submotifs are important for MAPK binding and docking activity (Bardwell, 2006; Bardwell & Bardwell, 2015). In the other invertebrate model organisms examined, however, the hydrophobic submotif was lacking one or both hydrophobic residues, suggesting it may no longer function as a high-efficiency MAPK docking site in these species. The SP102 site is pan-chordate, and is also present in most invertebrate phyla, including two of the most ancient—Porifera (sponges) and Cnidaria (sea anemones)—although in the three worm phyla and mollusks the proline has been lost. The SUFU-binding motif was, as expected, widely conserved. Finally, a site that may be equivalent to the SP130 site was also present in most of the species we examined; only in cephalochordates did we fail to find an SP within 10 residues of the SUFU-binding motif. To summarize, a functional D-site, SP102 site and SP130 site are present in GLI orthologs in most major animal phyla, whereas the SP116 site in GLI1 appears to be an innovation of placental mammals.

Interestingly, the R100 and S105 residues that we identified as being important for SUFU binding from scanning human tumor-derived GLI1 mutations (Fig 8) were also extremely well-conserved in all chordate and all invertebrate species that we examined, as was L106 (Fig 9). This suggests that these and neighboring residues may constitute an evolutionarily conserved region that functions to modulate the affinity of SUFU binding.

# A

| | D-site | SP102 | SP116 | SUFU-binding | SP130 |
|---|---|---|---|---|---|
| Placental Mammals | KKRALSI | VIRTSPSSL | SRCTₐSPGG | SYGHLS | IGTMSPSLGF |
| Marsupials | KKRALSI | VIRTSPSSL | SsCApPGG | SYGHLS | IGTiSPSLGF |
| Birds | KKRALSI | VIRTSPnSL | SRCASaGᴳG | SYGHLS | IsTiSPSLGy |
| Reptiles | KKRALSI | VIRTSPnSL | SRCASaGG | SYGHLS | IGavSPSLGF |
| Amphibians | KKRALᴸₘSI | VₘIRTSPnSL | SRCsSasG | SYGHLS | IGTiSPSLGy |
| Fish | KKRALSI | VIRTSPnSL | SRCnpnGa | SYGHLS | vsaMSPSLGy |

# B

| | D-site | SP102 | SP116 | SUFU-binding | SP130 |
|---|---|---|---|---|---|
| human GLI1 | KKRALSI | VIRTSPSSL | SRCTSPGG | SYGHLS | IGTMSPSLGF |
| human GLI2 | RKRALSI | MIRTSPNSL | – | SYGHLS | AGALSPAFTF |
| human GLI3 | RKRALSI | MIRTSPNSL | – | SYGHLS | ASAISPALSF |
| Tunicates (Ci) | KKRpLSI | MIRTSPSSL | – | SYGHLa | AGgISPtFGy |
| Cephalochordata (Bf) | RKRALSI | MIRTSPNSL | – | SYGHLS | – |
| Echinoderms (Sp) | KKRALSI | MIRTSPNSL | – | SYGHLS | AGTvSPvnwn |
| Arthropods (Dv) | KKRALSs | MIRfSPNSL | – | SYGHLS | ASAISPmaha |
| Annelids (Pd) | RKRALSh | ltRsSggSL | – | SYGHLS | AasfgagSPm |
| Molluscs (Ob) | RKRALSh | ltRsSegSL | – | SYGHLS | AasfgavSPa |
| Nematoda (Tp) | RKRpLSs | alRsStsSL | – | SvGHns | vSeSPtAFSP |
| Platyhelminthes (Sj) | KKRshSq | ltRsSqgSL | – | SYGHLS | AasLgaSPGt |
| Cnidaria (Nv) | RKRArSn | lIRnSPdSL | – | SfGHLS | PigfctSPac |
| Porifera (Aq) | KKRpLSI | lvRgSPtSL | – | SfGHLS | PSlytgttSr |

**Figure 9. Evolutionary conservation of phosphorylation sites near the SUFU-binding motif in GLI orthologs.**
**(A)** Alignment of GLI orthologs in vertebrates, focused on the five motifs indicated. In all cases, the motifs were found in the order shown, within a 60-residue region, N terminal to the zinc finger DNA-binding domain. The sequences shown represent a consensus from each vertebrate class; lowercase letters denote differences from the placental mammal consensus. The basic submotif of the D-site is colored blue, the hydrophobic residues of the D-site are in red, SP sites are pink, the SUFU-binding motif is green, and R100 and S105 are cyan. **(B)** Alignment of GLI orthologs in the two non-vertebrate chordate subphyla (tunicates and cephalochordates) and the major invertebrate phyla. All motifs were found in the order shown, within a 65-residue region, N-terminal to the zinc finger DNA-binding domain. Residue color scheme is as in 8A. Sequences shown were taken from the following representative species: Ci, *Ciona intestinales*, (accession number NP_001071951); Bf, *Branchiostoma floridae* (XP_035700250); Sp, *Strongylocentrotus purpuratis* (XP_030856399); Dv, *Drosophila virilis* (XP_015024014); Pd, *Platynereis dumerilii* (ADK38672); Ob, *Octopus bimaculoides* (XP_014767791); Tp, *Trichinella pseudospiralis* (KRY76744); Sj; *Schistosoma japonicum* (TNN10035); Nv, *Nematostella vectensis* (XP_048579969); and Aq, *Amphimedon queenslandica* (XP_003387859). Lowercase letters indicate residues that differ from the three human GLI proteins. A dash indicates that no SP site was found within 10 residues N terminal (SP116 site) or C terminal (SP130 site) to the SUFU-binding motif.

## Substitution of phosphorylated residues alters GLI1 transcriptional activity in cells

To begin to investigate the physiological consequences of phosphorylation of GLI1 at S102, S116 and S130, we constructed two mutant derivatives of full-length (FL) GLI1 protein. The first mutant, FL-GLI1$^{AAA}$, contains serine-to-alanine substitutions at those three residues and is therefore phosphodeficient. The second mutant, FL-GLI1$^{EEE}$, contains phosphomimetic serine-to-glutamate substitutions at those three residues; in Fig 7E, we showed that this mutant exhibits diminished SUFU-binding in vitro. Both variants were subcloned into the pCMV6-XL5 mammalian expression vector, resulting in constructs which drive the expression of full-length GLI1 cDNAs from the CMV promoter. These variants were then transiently transfected into Cos-1 cells, along with a luciferase reporter gene containing 12 copies of the consensus Gli-binding site (GACCACCCA) linked to a basal promoter (Kogerman et al, 1999). As shown in Fig 10A, vector-driven expression of wild-type GLI1 strongly activated the luciferase reporter in this system. Furthermore, compared with wild-type, the phosphomimetic mutant

GLI1$^{EEE}$ was hyperactive, driving a more than twofold increase in reporter activity. Moreover, this increase was statistically significant ($P < 0.01$). In contrast, the GLI1$^{AAA}$ mutant displayed a slightly reduced ability to drive reporter expression compared with wild type, although this difference was not statistically significant. Immunoblot analysis showed that expression of all three proteins was roughly equivalent (Fig 10A). To summarize, these data show that phosphomimetic mutations in S102, S116 and S130 result in an increase in GLI1-driven transcription.

To achieve stable expression of FL-GLI1 variants, we used the piggyBac transposon system to integrate the wild-type, AAA, and EEE alleles into the genome of NIH3T3 cells. In addition, as a control, the L124P allele (a substitution in the SUFU-binding site derived from a drug-resistant basal cell carcinoma tumor) was also expressed using this system. Stable transformants were then treated with a conditioned medium derived from cells expressing active Sonic Hedgehog ligand (Shh-N), or with a control conditioned medium lacking Shh-N. To assay Hh pathway activity, expression of the endogenous mouse *Gli1* locus was monitored by RT-PCR (Fig 10B). As mentioned in the introduction, the *Gli1* gene is a target of the Hh pathway; thus, expression of this locus is a measure of the level of activation of endogenous Hh pathway output.

The results were interesting for several reasons. First, the expression of the endogenous mouse *Gli1* locus, driven by transfected wild-type human GLI1 protein, was inducible (2.3-fold) by Shh-N, as expected. Second, the L124P allele was hyperactive (by 2.2-fold) in the absence of induction, consistent with its in vitro SUFU-binding defect (Fig 8). Interestingly, this allele was also inducible by Shh-N (1.6-fold), suggesting that L124P is not completely defective in SUFU binding in cells, and/or that there are additional mechanisms by which GLI1-driven transcription can be potentiated by Hh pathway stimulation that are independent of SUFU release. Notably, the phosphodeficient triple mutant S102A, S116A, S130A (GLI1$^{AAA}$) was slightly less active than wild-type GLI1 in the absence of Shh-N, and, importantly, was not significantly induced by ligand. This suggests that, in cells, phosphorylation of one or more of these three residues is necessary for full GLI1 activation. Finally, the phosphomimetic triple mutant S102E, S116E, S130E (GLI1$^{EEE}$) was not discernably different from wild-type GLI1 in the absence of Shh-N. Strikingly, however, transcription driven by GLI1$^{EEE}$ was substantially induced by ligand. Indeed, the level of induction exhibited by GLI1$^{EEE}$ was significantly greater than wild-type (4.3-fold for GLI1$^{EEE}$ versus 2.3-fold for wild-type GLI1). The lack of basal induction in the presence of GLI1$^{EEE}$ indicates that phosphorylation of S102, S116, and/or S130 is not sufficient to activate GLI1 in NIH3T3 cells. At the same time, the hyperinduction driven by GLI1$^{EEE}$ in the presence of Shh ligand indicates phosphorylation of S102, S116, and/or S130 can potentiate transcription driven by activated GLI1.

# Discussion

Given the widespread involvement of MAP kinase pathways in cancer, the more concentrated involvement of HH/GLI in many tumor types, and the substantial evidence of crosstalk between these pathways, it is important to understand the molecular mechanisms of this crosstalk. Indeed, such understanding could

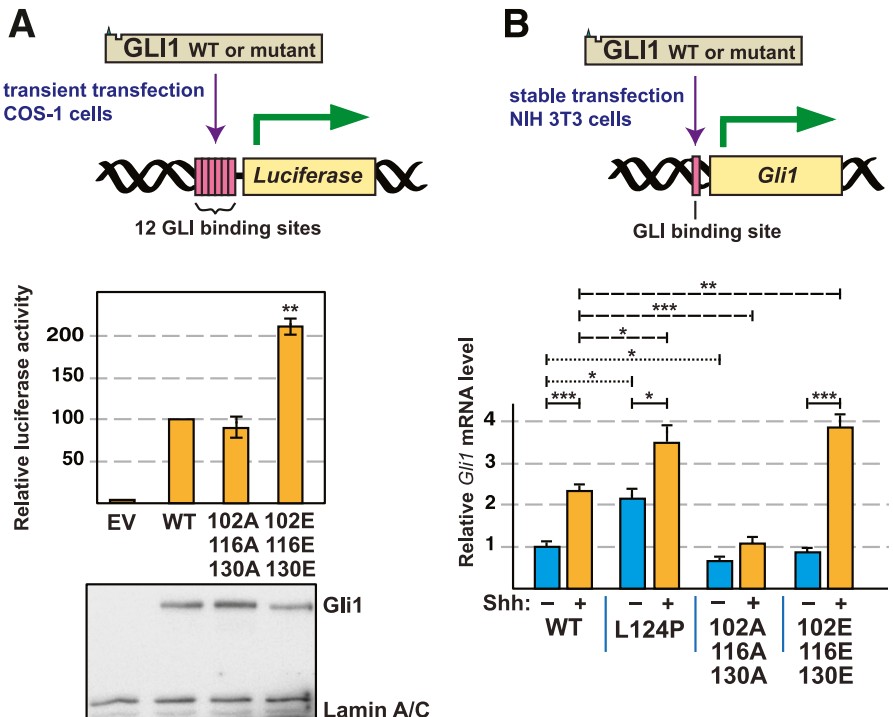

**Figure 10. Analysis of GLI1-driven transcription by GLI1 variants.**
**(A)** COS-1 cells were transfected with a luciferase reporter vector for GLI-driven transcription, TK-βgal (a control for transfection efficiency) and either pCMV6-FL-GLI1, pCMV6-FL-GLI1^AAA, or pCMV6-FL-GLI1^EEE, as indicated. Cells were collected 24 h later and assayed for reporter activity. Luciferase activity was first normalized to β-galactosidase and then normalized again so that wild-type activity equals 100 units. **(B)** NIH3T3 cells stably expressing full-length human GLI1 variants from piggyBac transposons were serum starved for 24 h and then treated with Shh-N conditioned media or control conditioned media. Expression of the endogenous mouse *Gli1* locus was analyzed by real-time quantitative RT-PCR. Fold change in mouse *Gli1* mRNA expression was measured using ΔΔCt analysis. mApple (expressed from the same transposon) served as an internal control. The data were further normalized so that uninduced wild type equals 1 unit. *P < 0.05, **P < 0.01, ***P < 0.001.

suggest new strategies for targeted intervention in cancer, and improve our understanding of signal crossover and network rewiring in tumor cells (Aberger & Ruiz i Altaba, 2014; Samatar & Poulikakos, 2014; Caunt et al, 2015; Pietrobono et al, 2019; Yesilkanal et al, 2021).

GLI1, GLI2, and GLI3 are the primary transcriptional effectors of the HH pathway. Responding to HH pathway signaling, GLI proteins regulate genes that promote self-renewal, proliferation, survival, and other processes. In recent years, however, evidence has accumulated indicating that GLI transcription factors receive inputs from many other pathways in the form of protein–protein interactions and post-translational modifications (Chen & Jiang, 2013; Montagnani & Stecca, 2019). These inputs presumably function to integrate the signaling status of the HH pathway and cross-regulating pathways into a coherent response.

Previously, we used a computational approach to identify a MAPK-docking site in the N-terminal portion of human GLI1, GLI2 and GLI3, and demonstrated that this D-site both bound to ERK2, and directed ERK2 to phosphorylate nearby target residues in GLI1 and GLI3. We also identified one of these sites as S343 in GLI3 and S130 in GLI1, both of which sit just C-terminal to the high-affinity binding site for SUFU (Whisenant et al, 2010). Here we took a biochemical approach to identify additional MAPK target sites, and to investigate the consequences of MAP kinase-mediated phosphorylation of GLI1 in this region, and presented five major findings.

First, we showed that GLI1 bound to SUFU with high affinity, and that in vitro phosphorylation of purified $GLI1_{68-232}$ by purified, active ERK2 weakened the affinity of GLI1–SUFU binding by almost 30-fold (Fig 2), an effect that we refer to as MAPK-mediated SUFU release. Second, we identified S102, S116 and S130 as the phosphorylation

sites in the N terminus of GLI1 that underlie this release, as we showed that phosphorylation of these three sites is both necessary and sufficient for this effect (Figs 3–6). Third, we provided evidence that multisite phosphorylation of these three sites is required for full SUFU release: phosphorylation of any one, or even any two, of the three sites did not result in the level of SUFU release seen when all three sites were able to be phosphorylated (Figs 5 and 6). These findings were corroborated when we analyzed a set of alleles containing phosphomimetic substitutions of S102, S116, and/or S130 in all possible combinations (Fig 7).

Our fourth major finding was the identification of a region of SUFU centered around Ser102 that plays a role in SUFU binding. We identified two GLI1 variants found in multiple patient-derived tumor samples, R100C and S105F, that substantially weakened SUFU binding (Fig 8), and have been highly conserved throughout animal evolution (Fig 9). Conceivably, R100, S102, and S105 may, like $^{120}$SYGHLS$^{125}$, make direct contact with SUFU. Alternatively, these residues may act indirectly, by stabilizing a GLI1 conformation that is accessible to SUFU. Of note, we also showed that the L124P substitution in GLI1, identified in a basal cell carcinoma that had acquired resistance to vismodegib, dramatically weakened binding to SUFU and potentiated GLI1-driven transcription, suggesting that this mutation may account for the drug resistance seen in the patient (Figs 8 and 10).

Finally, we showed that a mutant GLI1 allele containing phosphomimetic substitutions of S102, S116 and S130 displayed an increased ability to drive transcription of both a reporter gene and an authentic *Gli1* target gene. Conversely, substitution of S102, S116, and S130 with alanine (hence rendering them unphosphorylatable) resulted in a modest reduction in GLI1-driven transcription (Fig 10).

Taken together, our results suggest that phosphorylation of GLI1 on S102, S116, and/or S130, mediated by ERK2 or other kinases, weakens GLI1–SUFU binding, thereby facilitating GLI1 activation.

## MAPK phosphorylation occurs in an important regulatory region

For much of this study, we focused on a small region of GLI1, ~60 amino acids long, that contains both the [79]KKRALSI[85] D-site and the [120]SYGHLS[125] SUFU-binding motif. In this region, we characterized three ERK2-phosphorylation sites which cooperate to regulate SUFU binding. In addition, we found three tumor-derived mutations in this region—R100C, S105F, and L124P—that also disrupted SUFU binding. Interestingly, several other important functional motifs are also found in this region, including a degradation signal that spans the entire region (Huntzicker et al, 2006) and a ciliary localization signal that partially overlaps with the D-site and with the S70 and S86 MAPK target sites (Han et al, 2017). Other kinases also phosphorylate GLI1 in this region (Wang et al, 2012; Li et al, 2015; Schneider et al, 2015).

This ~60-residue region is well conserved between vertebrate and invertebrate GLI orthologs, as was first described by Croker et al (2006) who noted the homology in this region between *Drosophila Cubitis interruptus* (*Ci*), the founding *GLI* gene, and mouse and human GLI proteins (Croker et al, 2006). Here, we extended this observation to all the major animal phyla, particularly focusing on the conservation of the D-site, the SUFU-binding motif, and the SP102, SP116, and SP130 phosphosites (Fig 9). Whereas the SP116 site was highly conserved in placental mammals but not elsewhere, the other four sites were highly conserved in all vertebrates with minimal sequence changes, suggesting that MAPK-mediated regulation of GLI-SUFU binding may be widespread in vertebrates.

In fact, there are hints that such regulation may be present in some invertebrate organisms as well: A strong D-site (with consensus basic and hydrophobic submotifs) is present in chordates, sponges, and echinoderms. SP102 is present in these phyla as well as arthropods and anemones, and all invertebrate models we examined had an SP130-like site within 10 residues of the [120]SYGHLS[125] SUFU-binding motif. MAP kinase cascades are present in virtually all eukaryotic organisms, and thus predate the evolution of the Hedgehog signaling pathway, which is first found in animals, and presumed to have been absent from the last unicellular ancestor of animals (Ingham et al, 2011; Babonis & Martindale, 2017). Hence, MAPK regulation of GLI transcription factors could conceivably have evolved contemporaneously with the origination of the first *GLI* gene.

Three closely spaced residues—R100, S102, and S105—are strictly conserved in all chordate and invertebrate *GLI* orthologs we examined. Mutations in these residues, which are located about 20–25 residues N-terminal to the SYGHLS SUFU-binding motif, weakened SUFU binding affinity up to 35-fold (Fig 8). We propose that these and near-neighbor residues constitute a SUFU-affinity-modulating region that either contacts SUFU directly (i.e., orthosterically), or indirectly (i.e., allosterically) influences the ability of the SYGHLS motif to optimally engage with SUFU. Phosphorylation of this region on S102, whether by MAP kinases or other kinases, may be a mechanism of regulation of the GLI–SUFU interaction that is conserved throughout the animal kingdom.

The region from 1 to 232 in human GLI1 is not predicted to be intrinsically disordered by tools such as MobiDB (Piovesan et al, 2021). Nevertheless, the structure of the majority of this region cannot be predicted with confidence by Alphafold2 (Jumper et al, 2021; Varadi et al, 2022). Interestingly, the exception is a short region from L95 to N111, which is predicted by Alphafold2 to form two short antiparallel α helices connected by a β turn, with SP102 at the apex of this turn. The same structure is also predicted for the corresponding regions of human GLI2 and GLI3, consistent with the 79% sequence similarity between the three GLI proteins in this region. Although it is not immediately apparent how phosphorylation of S102 might alter this structure, it would clearly introduce a negatively charged phosphate in the vicinity of R100 and S105, two other residues that we showed are critical for SUFU binding.

## Multisite phosphorylation regulates SUFU binding

What is the mechanism by which phosphorylation of S102, S116, and S130 in GLI1 cooperate in SUFU release? Phosphorylation of these residues might sterically interfere with GLI1–SUFU binding. Alternatively, phosphorylation of these residues might initiate a structural rearrangement that results in the [120]SYGHLS[125] SUFU-binding motif being less accessible to SUFU. To gain further insight, it is necessary to understand how GLI1 SP102, SP116, and SP130 influence binding to SUFU both when phosphorylated and when unphosphorylated. Some hints come from structural studies of complexes of SUFU with GLI1-derived peptides. These studies showed that the [120]SYGHLS[125] motif binds directly (i.e., orthosterically) to SUFU by a β-strand addition mechanism, in which it combines with β-strands from both lobes of SUFU into a merged 3-stranded β-sheet (Cherry et al, 2013; Zhang et al, 2013). It is straightforward to envision how phosphorylation of the serines/tyrosine in this motif might block SUFU binding by steric clash and/or electrostatic repulsion. There is no evidence, however, that these residues are phosphorylated by any kinases, and our data show that they are not phosphorylated by ERK2. The GLI1 peptide that Zhang et al (2013) co-crystalized with SUFU spanned residues 112–128, and thus included SP116, but not SP102 or SP130. The GLI1 peptide used by Cherry et al (2013) spanned residues 115–131, and thus included both SP116 and SP130, but again not SP102. In both studies, however, the SP sites did not have visible electron density, suggesting that they did not make direct contact with SUFU. For this reason, we lean in favor of allosteric communication as the mechanism by which phosphorylation of S102, S116 and S130 promotes SUFU release (Nussinov & Tsai, 2013). In this regard, it should be noted that the GLI1 112–128 peptide used by Zhang et al (2013) bound to SUFU with micromolar affinity in isothermal titration calorimetry assays, and they obtained substantially better binding by using longer GLI1 peptides (e.g., 97–143) that included not only SP116 but also SP102 and SP130. This observation is compatible with the idea that residues around SP102 and SP130 play a role in SUFU binding; it is also compatible with either an orthosteric or allosteric influence of the three SP sites. Neither group examined the binding of phosphorylated versions of their peptides.

How might multisite phosphorylation-mediated allosteric inhibition of GLI1–SUFU binding work? To speculate on this question, it

is helpful to examine studies that tackle related questions for other proteins (Nishi et al, 2014). Conformational rearrangements may be initiated when phosphorylated residues form salt bridges or hydrogen bonds with residues elsewhere in the protein (Pearlman et al, 2011; Hunter, 2012); these rearrangements can propagate to nearby regions (Kumar et al, 2012). For example, in NFAT, multisite phosphorylation favors a conformation that occludes a nuclear-localization signal (Shen et al, 2005), and in 4E-BP2, multisite phosphorylation induces a rearrangement of an α helical-binding motif into a β sheet, thereby inhibiting binding (Bah et al, 2015). Phosphorylation can also modulate the binding of an auto-inhibitory domain, thereby occluding/revealing a binding site for another protein (Ferreon et al, 2009). At a more abstract level, phosphorylation results in a change in free energy that favors one conformation over others. Multiple such changes (resulting from multisite phosphorylation) are additive at the level of free energy, but multiplicative in terms of their effect on the equilibrium constant, according to standard thermodynamic principles.

### Multiple mechanisms of MAPK-mediated crosstalk

There are likely to be multiple mechanisms of crosstalk between the RAS/MAPK pathway and the HH pathway (Rovida & Stecca, 2015). This is certainly true if we consider the branched signaling pathways downstream of activated receptor tyrosine kinases or RAS (Eberl et al, 2012), but is true even if we only focus on effects downstream of MAP kinase activation. ERK1/2 have a large number of substrates, including many transcription factors such as Elk-1, Ets1/2, Fos, and Pax6, some of which could cooperate with GLI1 proteins to regulate particular subsets of target genes (Maik-Rachline et al, 2019). Active ERK1/2 also activate several down-stream kinases including MSK1/2 and pp90RSK (Morrison, 2012). These kinases could in turn regulate GLI proteins directly (by phosphorylating them), or indirectly (by phosphorylating other proteins). With respect to the latter possibility, activated pp90RSK has been shown to promote GLI2 stability by phosphorylating and thereby inhibiting GSK3β, which is a negative regulator of GLI stability (Liu et al, 2014). It seems likely that different mechanisms of crosstalk will predominate to a greater or lesser extent depending on the cell type and context, including the oncogenic load (Aberger & Ruiz i Altaba, 2014). In this regard, it is worth noting that a recent phosphoproteomic study of short-term changes in a HH-responsive human medulloblastoma cell line found that treatment with a HH agonist led to inhibition of ERK signaling, whereas treatment with vismodegib led activation of ERK signaling (Scheidt et al, 2020). The latter finding suggests how ERK-mediated crosstalk could initially maintain some GLI activity in the presence of vismodegib.

In this study, we focused on MAPK phosphorylation events in the N-terminus of GLI1, where the MAPK-docking site and SUFU-binding motif are located. We did not attempt to assess MAPK-mediated phosphorylation of residues distal to residue 232, although this might be a productive direction for future studies. There are no canonical MAPK target sites (SP or TP) in the DNA binding domain of GLI1 (residues 233–401). There are, however, a total of 12 SP and three TP sites in the remainder of the molecule.

Phosphomimetic substitutions of S102, S116 and S130 in full-length GLI1 did not lead to complete dissociation of SUFU, nor did mutations in the SUFU-binding motif (Fig 7E). This suggests that additional events may be required for full release of SUFU, and is consistent with evidence for a second SUFU-binding domain in the C-terminal portion of GLI1, GLI2, and fly Ci (Merchant et al, 2004; Han et al, 2015; Oh et al, 2015). SUFU, like GLI1, is subject to complex levels of regulation, and some of these might cooperate with GLI1 phosphorylation in the 75–135 region to promote full SUFU dissociation. In addition to binding to GLI1, GLI2, and GLI3, SUFU binds to several other proteins (Lin et al, 2014; Wu et al, 2017b), and is itself post translationally modified by phosphorylation and ubiquitylation (Chen et al, 2011; Raducu et al, 2016; Infante et al, 2018). Finally, we note that there is some evidence suggesting that GLI1 activation may lead to a structural rearrangement of the GLI1–SUFU complex, but not to complete dissociation of this complex (Zhang et al, 2017).

### Conclusion

Here we have shown that the ERK2 MAP kinase directly phosphorylates GLI1 on multiple evolutionarily conserved target sites near the SUFU-binding motif, thereby facilitating the release of SUFU from GLI1. We suggest that this MAPK-mediated SUFU release might be an important component of the molecular mechanism by which MAP kinases activate GLI1 in cancers and other pathological scenarios. We further suggest that phosphorylation of these residues may play a widespread role in Hedgehog signaling in animals.

# Materials and Methods

### Genes

The mammalian genes used in this study were human *GLI1* (NCBI accession number NM_005269) and human *SUFU* (NM_016169).

### Plasmids

The vector used for generating GST fusion proteins was pGEX-LB, a derivative of pGEX-4T-1. In pGEX-LB, an encoded Pro residue is replaced with a Gly-Gly-Gly-Gly-Gly-Ser-Gly coding sequence to promote the independent functioning of the GST and fusion moieties (Bardwell et al, 2001). Plasmid GST-GLI1(68–232), which encodes a fusion of GST to residues 68–232 of human GLI1, was previously described (Whisenant et al, 2010). Plasmid pGEM4Z-SUFU encodes full-length (residues 1–484) human SUFU downstream of a promoter for SP6 RNA polymerase. To construct this plasmid, the SUFU open reading frame was amplified by PCR using Open Biosystems clone 3533158 as the template and primers JB-SUFU-UP1 (5'-GCGGGATCCACCATGGCGGAGCTGCGGCCT) and JB-SUFU-DN1 (5'-GCGTCGACTTAG TGTAGCGGACTGTCGAACA). The PCR product was digested with the *BamH*I and *Sal*I restriction enzymes and inserted into pGEM4Z (Promega) that had been digested with the same enzymes. Plasmid GST-SUFU encodes a fusion of GST to full-length

human SUFU. To construct this plasmid, the *BamH*I-SUFU-*Sal*I cassette described above was inserted into the corresponding sites of plasmid pGEXLB (Bardwell et al, 2001). Quikchange (Invitrogen) was used for site-directed mutagenesis; all mutations were confirmed by sequencing. For the experiments shown in Fig 10A, we obtained a clone expressing full-length (residues 1–1,106) untagged GLI1 in the pCMV6-XL5 vector backbone from OriGene Technologies (Cat. no. SC125780). For Fig 10B, we obtained a PiggyBac mApple fusion cassette derived from the EF1 constitutive active expression system PB-EF1-MCS-IRES-Neo (PB533A-2; Systems Biosciences) where mApple cDNA was inserted between *Nhe*I/*Eco*RI sites and the *GLI1* gene or its variants inserted into *Eco*RI downstream of mApple using the InFusion cloning kit (Takara Bio). Quikchange (Invitrogen) was used for site-directed mutagenesis; all mutations were confirmed by sequencing.

## Protein purification

GST fusion proteins were expressed in bacteria, purified by affinity chromatography using glutathione-Sepharose (GE Healthcare), eluted from the matrix by reduced glutathione, dialyzed to remove the glutathione, quantified by Bradford assay and NanoDrop analysis, and flash frozen in aliquots at 80°C, as described elsewhere (Bardwell et al, 2001; Whisenant et al, 2010; Gordon et al, 2013).

## In vitro transcription and translation

Proteins labeled with [$^{35}$S]-methionine were produced by coupled transcription and translation reactions (SP6 or T7, Promega). Translation products were partially purified by ammonium sulfate precipitation (Bardwell & Shah, 2006), and resuspended in "Binding Buffer" (20 mM Tris–HCl, pH 7.5, 125 mM KOAc, 0.5 mM EDTA, 5 mM DTT, 0.1% [vol/vol] Tween20, and 12.5% [vol/vol] glycerol) before use in binding assays.

## Protein kinase assays

The protein kinase assay shown in Fig 4 was performed as described elsewhere (Bardwell & Bardwell, 2015). The buffer for kinase assays was MAP kinase buffer (50 mM Tris–HCl [pH 7.5], 10 mM MgCl$_2$, 1 mM EGTA, and 2 mM DTT) with 50 $\mu$M ATP and 1 $\mu$Ci of [$\gamma$-$^{32}$P]-ATP.

High efficiency phosphorylation of GST-GLI1$_{68-232}$ for use in SUFU binding assays (Figs 2, 3, 5, and 6) was accomplished by incubating 1 $\mu$M (~1.8 $\mu$g) GST-GLI1$_{68-232}$ with 100 units (~10 ng) of purified active ERK2 for 2 h at 30°C in MAP kinase buffer supplemented with 400 $\mu$M ATP and 1 mg/ml molecular biology grade BSA. Reactions were then stopped by the addition of an excess volume of ice cold 1× Binding Buffer (described above) in which the EDTA had been raised to 10 mM. Mock-treated samples were treated identically except no ERK2 was added. The phosphorylated or mock-treated GST-GLI11$_{68-232}$ was then aliquoted to tubes at the appropriate concentration for the binding isotherm (final concentration 1.25–80 nM). Then, glutathione-Sepharose beads (20 $\mu$l of a 50% slurry) were added, the samples were rocked for 1 h at room temperature, and the resulting bead-bound protein complexes were isolated by sedimentation and washed thoroughly in binding buffer. This step

served to remove the ERK2 and BSA from the previous step, exchange MAP kinase buffer for binding buffer, and to bind the GST-GLI1 molecules to the beads. After that, $^{35}$S-radiolabeled full-length human SUFU protein was added as described below.

For the kinase assays shown in Figs 2B and 3B, aliquots of the high-efficiency phosphorylation reactions described above were removed at time 0 and spiked with 1 $\mu$Ci of [$\gamma$-$^{32}$P]-ATP. These reactions were incubated at 30°C for 2 h or the time shown, whereupon they were stopped by the addition of SDS sample buffer. Fig 2B contains additional controls; the "no ATP" samples were not spiked with [$\gamma$-$^{32}$P]-ATP, nor did they contain any "cold" ATP.

## Protein binding assays

For the experiments shown in Figs 6 and 7, $^{35}$S radiolabeled GLI1 protein was prepared by in vitro translation and partially purified by ammonium sulfate precipitation. Approximately 1 pmol of $^{35}$S-GLI1 was added to each 200 $\mu$l binding reaction; 10% of this amount was loaded in the "Input" lane. Purified GST-SUFU was added at concentrations that varied from 1.25 to 80 nM. The buffer for binding reactions was Binding Buffer (described above) to which 1 mg/ml molecular-biology grade BSA was added to block nonspecific protein–protein or protein-bead interactions. The binding reactions were incubated for 15 min at 30°C. Then, glutathione-Sepharose beads (20 $\mu$l of a 50% slurry) were added, the samples were rocked for 1 h at room temperature, and the resulting bead-bound protein complexes were isolated by sedimentation, washed thoroughly with binding buffer to remove unbound protein, and resolved by 10% SDS–PAGE on the same gel. Other details of the protein binding assays are as described elsewhere (Bardwell et al, 2001; Gordon et al, 2013).

For the experiments shown in Figs 2, 3, and 5, $^{35}$S radiolabeled full-length human SUFU protein was prepared by in vitro translation and partially purified by ammonium sulfate precipitation. Approximately 1 pmol of $^{35}$S-SUFU was added to each 200 $\mu$l binding reaction; 10% of this amount was loaded in the "Input" lane. The phosphorylated and mock-treated GST-GLI1$_{68-232}$ used in these assays was prepared as described above (see "Protein kinase assays"). The buffer for binding reactions was Binding Buffer + BSA, as described above. The binding reactions were incubated for 15 min at 30°C, then rocked for 1 h at room temperature. The resulting bead-bound protein complexes were isolated by sedimentation, washed thoroughly with binding buffer to remove unbound protein, and resolved by 10% SDS–PAGE on the same gel.

## Statistical analysis and fitting

Statistical analysis of binding assay results was performed using Welch's unequal variance *t* test with two tails. The Bonferroni correction factors were obtained by dividing the standard *P*-value threshold of 0.05 by the number of planned comparisons. The Bonferroni correction addresses the following issue: in the absence of a correction for multiple hypothesis testing, if one were to test 20 hypotheses, all of which were false, with a 95% confidence threshold, the probability that one or more of them would appear true just by chance would be 1–0.95$^{20}$, or 0.64. With the Bonferroni correction, the

probability of one or more false positives occurring in the set of 20 comparisons is reduced to $1-0.9975^{20}$, which is less than 0.05.

All differences that were found to be significant by the Bonferroni-corrected Welch's *t* test were also found to be significant using Dunnett's T3 multiple comparisons test. Graphpad Prism and Microsoft Excel software packages were used for statistical analysis.

Binding isotherm analysis was conducted as follows: Emax and Kd for the wild-type protein were determined by fitting wild-type data from each experiment to the formula:

$$\% = Emax \frac{x}{Kd + x}.$$

where "%" is the percent of the radiolabeled protein bound, and *x* is the concentration of the GST-GLI1$_{68-223}$ or GST-SUFU (i.e., 1.25, 2.5, 5, 10, 20, 40, or 80 nM). The Emax for wild type was then used to normalize the wild type and all mutant variants analyzed in the same experiment.

Kd values for each concentration were determined from the formula:

$$Kd = x \frac{(100 - \%)}{\%}.$$

where "%" and *x* are as defined above. The set of Kd values so obtained was used as the input array for the *t* test.

### Sequencing of *GLI1* alleles from vismodegib-resistant basal cell carcinoma samples

Five to eight 10-$\mu$m sections were obtained from the formalin-fixed, paraffin-embedded (FFPE) tumor block of resistant BCC tumors, and DNA was isolated using the QIAGEN DNeasy blood and tissue kit according to manufacturer's protocol (QIAGEN). The exonic regions of *Gli1* were amplified using the Access Array platform (Fluidigm). The samples were amplified in a multiplex format with genomic DNA (100 ng) according to the manufacturer's recommendation (Ambry Genetics). Subsequently, the multiplexed library pools were subjected to deep sequencing using the Illumina MiSeq platform. After demultiplexing and FASTQ file generation for the raw data, 150-base pair reads were aligned to the human reference genome sequence (hg19) using the BWA aligner. SAMtools mpileup was used to call variants. Calls required a minimum allele frequency of 5% at a position with a read depth of >100. Identified variants were annotated using SeattleSeq138 to exclude nonpathogenic variants reported in dbSNP138 and to identify variants that had non-synonymous consequences or affected splice sites.

### Sequence homology analysis

For the analysis shown in Fig 9A, we examined more than 300 sequences from the US National Center for Biotechnology Information (NCBI) vertebrate orthologs for GLI1 https://www.ncbi.nlm.nih.gov/gene/2735/ortholog/?scope=7776. Predicted low quality sequences were omitted. For the analysis shown in Fig 9B, we BLASTed the sequence of human GLI1 residues 75–135 against nonredundant protein sequences from the phylum indicated, using the *Choose Search Set* section of the BLAST form to limit the search to individual phyla or subphyla (Johnson et al, 2008).

### Cell based assays

Luciferase assays were performed essentially as described previously (Sprowl-Tanio et al, 2016). COS-1 cells were transfected with 0.5 $\mu$g 12GLI-RE-TKO-luc (Kogerman et al, 1999), 0.5 $\mu$g pCMV6-FL-GLI1 plasmid, and 0.1 $\mu$g thymidine kinase $\beta$-galactosidase plasmid using a lipofectin-based transfection reagent. Cells were harvested 24-h post transfection and assayed for luciferase activity and $\beta$-galactosidase activity (used for normalization). Immunoblot analysis was also as described (Sprowl-Tanio et al, 2016); the antibodies used were rabbit monoclonal anti-GLI1 (Clone EPR4523; Origene Technologies) at a 1:1,000 dilution and anti-Lamin A/C (2032; Cell Signaling Technologies) at 1:1,000.

Stable cells lines were generated using NIH3T3 cells and piggyBac transposons containing coding sequences for mApple and full-length human *GLI1*. Cells were transfected with PEI (NC1038561; Thermo Fisher Scientific) per the manufacturer's protocol and then selected for using 500 $\mu$g/ml Geneticin G-418 (50841720; Thermo Fisher Scientific) until all non-transfected cells were no longer viable. NIH3T3 cell lines were plated to confluency, serum starved, and treated (or mock-treated) with Shh-N-conditioned media (1: 100) for 24 h. RNA was isolated using the Directzol RNA MiniPrep Plus (ZYMO Research). Real-time RT-quantitative (q)PCR was performed using the iTaq Univer SYBR Green 1-Step Kit (Bio-Rad) on a StepOnePlus Real-time PCR system (Applied BioSystems) using primers for *Gli1* (forward: 5′-GCAGGTGTGAGGCC AGGTAG TGACGA TG-3′, reverse: 5′-CGCGGG CAGCAC TGAGGA CTTGTC-3′) and *mApple* (forward: 5′-ACCTAC AAGGCC AAGAAG CC-3′, reverse: 5′-GCGTTC GTACTG TTCCAC GA-3′). Fold change in *Gli1* mRNA expression was measured using $\Delta\Delta$Ct analysis with *mApple* as an internal control. Experiments were repeated three times, each experiment had triplicate technical replicates.

## Supplementary Information

## Acknowledgements

This work was supported by National Institute of General Medical Sciences research grant P50GM76516, National Cancer Institute (NCI) award P30CA062203, and UC Cancer Research Coordinating Committee awards CRR-12-201302 and CTR-20-637218. ML Waterman and B Wu were also supported by NCI grants CA096878, CA108697, and U54CA217378. SX Atwood was also supported by NCI grant CA237563. KY Sarin is the DG "Mitch" Mitchell Clinical Investigator, and is supported by the Damon Runyon Cancer Research Foundation (CI-104-19) and NCI grant CA211793.

### Author Contributions

AJ Bardwell: conceptualization, formal analysis, supervision, validation, investigation, visualization, methodology, and writing—original draft, review, and editing.
B Wu: formal analysis and investigation.

KY Sarin: conceptualization, resources, data curation, formal analysis, funding acquisition, and investigation.

ML Waterman: conceptualization, resources, supervision, and funding acquisition.

SX Atwood: conceptualization, resources, formal analysis, supervision, funding acquisition, and investigation.

L Bardwell: conceptualization, formal analysis, supervision, funding acquisition, validation, investigation, visualization, methodology, project administration, and writing—original draft, review, and editing.

## Conflict of Interest Statement

The authors declare that they have no conflict of interest.

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
