## [Reviewer comments · Life Science Alliance]

Life Science Alliance

ERK2 MAP kinase regulates SUFU binding by multisite phosphorylation of GLI1

Andrea Bardwell, Beibei Wu, Kavita Sarin, Marian Waterman, Scott Atwood, and Lee Bardwell

DOI: <https://doi.org/10.26508/lsa.202101353>

Corresponding author(s): Lee Bardwell, University of California, Irvine

Review Timeline:

Submission Date:	2021-12-24
Editorial Decision:	2022-02-15
Revision Received:	2022-05-10
Editorial Decision:	2022-06-03
Revision Received:	2022-06-17
Accepted:	2022-06-21

Scientific Editor: Novella Guidi

Transaction Report:

February 15, 2022

Re: Life Science Alliance manuscript #LSA-2021-01353-T

Dr. Lee Bardwell
University of California, Irvine
Developmental & Cell Biology
2208 Natural Sciences I
University of California
Irvine, CA 92697-2300

Dear Dr. Bardwell,

Thank you for submitting your manuscript entitled "ERK2 MAP kinase regulates SUFU binding by multisite phosphorylation of GLI1" to Life Science Alliance. The manuscript was assessed by expert reviewers, whose comments are appended to this letter. We, thus, encourage you to submit a revised version of the manuscript back to LSA that responds to all of the reviewers' points.

Thank you for this interesting contribution to Life Science Alliance. We are looking forward to receiving your revised manuscript.

Sincerely,

B. MANUSCRIPT ORGANIZATION AND FORMATTING:

Reviewer #1 (Comments to the Authors (Required)):

In this manuscript Bardwell and colleagues investigated the cross-talk between Hedgehog pathway and MAPK signaling, which is the major responsible for the emergence of clinical resistance to Hedgehog pathway inhibitors. They show that ERK2 MAP kinase regulates SUFU binding by multisite phosphorylation of GLI1. Authors nicely demonstrate that MAP kinase ERK2 phosphorylates GLI1 on three evolutionarily-conserved target sites (S102, S116 and S130) located near the high-affinity binding site for the negative regulator SUFU. These phosphorylation events cooperate to weaken the affinity of GLI1-SUFU binding and, thus, enhance GLI1 activity. Indeed, GLI1 mutants containing phosphomimetic substitutions of S102, S116 and S130 display an increased ability to drive transcription in Cos-7 and NIH3T3 cells, well known HH signaling competent cell lines.

This is a well-presented body of work with excellent protein kinase and binding assays, and functional analyses in mammalian cells. The manuscript is well written and easy to follow. All the experiments are professionally executed. This work is of high interest in the Hedgehog field and for the readership of Life Science Alliance, as it unravels a potential mechanism by which phosphorylation of S102, S112 and S130 cooperate in SUFU release, one of the most important negative regulator of GLI1.

Specific comment:

Luciferase assay clearly shows that the phosphomimetic triple mutant S102, S116 and S130 (GLI1-EEE) increases the transcriptional activity compared to GLI1 wild-type. It would be nice to see the functional effect of the triple mutants GLI1-EEE and GLI1-AAA (compared to wild-type GLI1) in mammalian cell growth. Does the increased transcriptional activity result in increased cell proliferation?

Minor points:

-In Fig. 7C (legend), replace "no effect" with "no effect".

Reviewer #2 (Comments to the Authors (Required)):

The manuscript by Bardwell et al., reveals a mechanism by which MAPK signaling (specifically ERK1/2), promotes the activity of the Gli1 transcription factor via limiting its interaction with its negative regulator SUFU. Using a series of Gli1 mutants, the authors identify three key residues (S102, S116 and S130) that are phosphorylated by ERK1/2. Multi-site ERK1/2-mediated phosphorylation of Gli1 diminishes its binding to SUFU, thereby facilitating Gli1 activation.

This is a well designed and executed study, and the conclusions that are drawn are supported by the data.

1. What was the rationale for using a "mock treated" control for the binding studies, rather than a kinase-dead ERK?
2. Are the GST-Gli1 WT and mutant proteins intact, particularly the phosphorylated recombinant proteins? A Western blot using GST and/or N-terminal Gli1 antibodies would confirm intact expression and eliminate the degradation of Gli1 (and mutants thereof) as a contributing factor to its decreased interaction with SUFU.

Minor comments

Fig 2B: the "kDa" is missing from the second gel panel.

Fig 3B/C and 4C and 6C and 9A: the "kDa" is missing from all gels.

Reviewer #3 (Comments to the Authors (Required)):

Summary

The major claims are that ERK2 phosphorylates Gli1 and when three serines residues (102, 116 and 130) are all phosphorylated, the affinity of Sufu, the Gli repressor is decreased. Tumour-derived mutations which are located nearby also diminish Sufu binding affinity. This provides a potential mechanism for disease in this case. It has also been demonstrated (by mimetics) that phosphorylation of these residues results in increased Gli transcription factor activity.

This is a significant advance in understanding how Hedgehog signalling is activated and can become overactivated in malignant

growth.

Main points

1) ERK2 phosphorylation of GLI1 modulates GLI1-SUFU binding

Data are strongly supportive. Note, there is a control with GST only shown in the Figures, but I think that the text should mention this to prove that binding is not GST-mediated.

2) Phosphorylation occurs on canonical sites and is necessary for binding modulation

Data are strongly supportive.

3) Ser102, Ser116 and Ser130 are efficiently phosphorylated by ERK2.

Data are strongly supportive. However, phosphorylation of S201 is similar to S102. I would like an explanation of why S201 wasn't pursued in the Sufu release studies.

4) Phosphorylation of S102, S116 and/or S130 are necessary and sufficient for SUFU release, residues act cooperatively and have the same effect in full-length Gli. Mutations found in tumours reduce binding to Sufu.

Data are supportive. With regards to all mutant experiments, I would like to see data that the mutant proteins are as well expressed as WT and/or that the same amount of protein was used in binding experiments. This could be a supplementary figure.

5) Phosphorylation sites in Gli affect transcriptional activity.

Data is strongly supportive.

Additional Points

Scheidt et al. 2020 performed an interesting phosphoproteomic study which should be referenced and discussed. Their data seem to demonstrate an opposite effect to this manuscript, that ERK/MAPK signaling increases with inhibition of the Hh signalling .

Another interesting point for discussion is that in the predicted GLI1 structure produced by AlphaFold, residues 95-111 are predicted to be alpha-helical. One of the phosphorylated residues in this manuscript is contained within this helix. It would be interesting to discuss whether the presence of a helix here would effect phosphorylation and Sufu binding/release.

Bardwell et al., ERK2 MAP kinase regulates SUFU binding by multisite phosphorylation of GLI1**Response to Reviewers**

We thank all three reviewers for their useful and thoughtful comments.

Reviewer #1 (Comments to the Authors (Required)):

In this manuscript Bardwell and colleagues investigated the cross-talk between Hedgehog pathway and MAPK signaling, which is the major responsible for the emergence of clinical resistance to Hedgehog pathway inhibitors. They show that ERK2 MAP kinase regulates SUFU binding by multisite phosphorylation of GLI1. Authors nicely demonstrate that MAP kinase ERK2 phosphorylates GLI1 on three evolutionarily-conserved target sites (S102, S116 and S130) located near the high-affinity binding site for the negative regulator SUFU. These phosphorylation events cooperate to weaken the affinity of GLI1-SUFU binding and, thus, enhance GLI1 activity. Indeed, GLI1 mutants containing phosphomimetic substitutions of S102, S116 and S130 display an increased ability to drive transcription in Cos-7 and NIH3T3 cells, well known HH signaling competent cell lines.

This is a well-presented body of work with excellent protein kinase and binding assays, and functional analyses in mammalian cells. The manuscript is well written and easy to follow. All the experiments are professionally executed. This work is of high interest in the Hedgehog field and for the readership of Life Science Alliance, as it unravels a potential mechanism by which phosphorylation of S102, S112 and S130 cooperate in SUFU release, one of the most important negative regulator of GLI1.

Response: We are very gratified that Reviewer #1 found our study to be “of high interest in the Hedgehog field and for the readership of Life Science Alliance” with “excellent protein kinase and binding assays”, and other “professionally executed” experiments. We are also very pleased that the reviewer found our manuscript to be “well written and easy to follow”.

Specific comment:

Luciferase assay clearly shows that the phosphomimetic triple mutant S102, S116 and S130 (GLI1-EEE) increases the transcriptional activity compared to GLI1 wild-type. It would be nice to see the functional effect of the triple mutants GLI1-EEE and GLI1-AAA (compared to wild-type GLI1) in mammalian cell growth. Does the increased transcriptional activity result in increased cell proliferation?

Response: The experiment suggested by the reviewer would clearly be very interesting to perform, and is a critical question for us to ask in the future. However, we are reluctant to perform such studies at this time for two reasons. First, we believe that there are likely to be additional mechanisms of MAPK-mediated crosstalk with GLI1, such as regulation of GLI1 protein stability, for which the relevant phosphorylation sites remain to be identified. Second, our current cell-based assays are not set up to address proliferative effects of GLI1 variants with the level of rigor that we would desire. The next stage of our study will involve CRISPR-based mutagenesis to construct endogenous alleles in non-transformed human cells of the three sites characterized in this manuscript, plus additional sites that we may still need to identify.

Minor points:

-In Fig. 7C (legend), replace "no effect" with "no effect".

Response: We thank the Reviewer for catching this typo, which we have corrected.

Reviewer #2 (Comments to the Authors (Required)):

The manuscript by Bardwell et al., reveals a mechanism by which MAPK signaling (specifically ERK1/2), promotes the activity of the Gli1 transcription factor via limiting its interaction with its negative regulator SUFU. Using a series of Gli1 mutants, the authors identify three key residues (S102, S116 and S130) that are phosphorylated by ERK1/2. Multi-site ERK1/2-mediated phosphorylation of Gli1 diminishes its binding to SUFU, thereby facilitating Gli1 activation.

This is a well-designed and executed study, and the conclusions that are drawn are supported by the data.

Response: We are very pleased that Reviewer #2 found our study to be “well-designed and executed” and that “the conclusions that are drawn are supported by the data.” We thank the reviewer for these supportive comments.

1. What was the rationale for using a "mock treated" control for the binding studies, rather than a kinase-dead ERK?

Response: We thank the reviewer for raising this important point.

The active ERK2 kinase that we used in these experiments is a highly-active preparation that is sold by New England Biolabs. This commercial enzyme preparation has been used for many years in experiments conducted in our laboratory, as well as many other laboratories. This preparation has consistently been at least as active as the ERK2 that we were able to produce in our own laboratory. However, New England Biolabs does not sell a corresponding kinase-dead preparation.

Our mock treated samples were identical in every way (all buffer components including Mg²⁺⁺, ATP, glycerol, BSA, etc. etc.) to the ERK2-treated samples except for the presence of the ERK2 enzyme preparation, which was a small fraction of the total volume. The ERK2 is then purified away in a step in which we rebind the GST-GLI1 to glutathione Sepharose beads. Finally we should note that our binding assay does not contain Mg²⁺⁺ or ATP; thus, even if a trace amount of ERK2 remains in the binding assay, it will not be able to phosphorylate anything.

Further evidence that ERK2-mediated SUFU release is due to ERK2-mediated phosphorylation, and not due to (a) some other effect of ERK2 (e.g. competitive binding) or some other activity in the ERK2 preparation (e.g. a protease) are:

- i. There was no sign of protein degradation during phosphorylation of GST-GLI or during the binding assays, This can be seen in the new supplemental Figures S1, S2, S3, and S5, as well as in the new Fig 2B (for more detail on this point, please see the response to this Reviewer's point 2, below).
- ii. ³⁵S-labelled SUFU was also clearly not degraded during the binding assays, as all binding assays were analyzed by SDS PAGE, and included a lane of SUFU (the 'Input' sample in Figs. 2 and 3) that was not used in a binding assay point. The electrophoretic mobility

- of SUFU in binding assay samples was identical to the electrophoretic mobility of the input SUFU, indicating no degradation.
- iii. ERK2-mediated SUFU release was abolished by mutation of S102, S116 and S130 in GLI1 to alanine (Fig 5), the same three residues that are the most prominently phosphorylated by ERK2 (Fig. 4).
 - iv. The effect of ERK2-treatment of GLI1 on SUFU release was mimicked by phosphomimetic substitution of these same three residues in GLI1 (Fig. 6).

2. Are the GST-Gli1 WT and mutant proteins intact, particularly the phosphorylated recombinant proteins? A Western blot using GST and/or N-terminal Gli1 antibodies would confirm intact expression and eliminate the degradation of Gli1 (and mutants thereof) as a contributing factor to its decreased interaction with SUFU.

Response: The reviewer raises an important concern. In the revised manuscript, we have added several new supplemental figures, as well as a new Fig 2B, in order to show (1) that GST-GLI1 and its mutant variants are not degraded during the phosphorylation procedure; (2) That GST-GLI1 and its mutant variants are not degraded during the binding assay procedure; (3) that there was equal loading of GST-GLI1 and its mutant variants in the binding assays:

1. Figure 2 has been revised with the addition of the new Fig 2B. This figure shows that phosphorylation of wild-type GST-GLI1 is not accompanied by any gel sign of degradation.
2. A new supplemental Figure S1 has been added. This figure shows representative binding assay gels from the experiments shown in Figure 2. The position of mock treated and phosphorylated GST-GLI1 is indicated, and it is evident that there is no sign of degradation during the binding assay procedure.
3. A new supplemental Figure S2 has been added. This figure shows binding assay gels from the experiments shown in Figure 3. The position of mock treated and phosphorylated GST-GLI1 and GST-GLI1^{7A} is indicated, and it is evident that there is no sign of degradation during the binding assay procedure.
4. A sentence has been added to the Fig 4C legend to clarify that all the variants shown in that figure migrated at the expected molecular weight for GST-GLI1₆₈₋₂₃₂ (44 kDa) following phosphorylation.
5. A new supplemental Figure S3 has been added. This figure shows representative binding assay gels from the experiments shown in Figure 5. The positions of mock treated and phosphorylated GST-GLI1 are indicated, and it is evident from comparing the mobility of the variants to the trace amount of BSA in each lane that there is no sign of degradation during the binding assay procedure.

Minor comments

Fig 2B: the "kDa" is missing from the second gel panel.

Response: This has been corrected.

Fig 3B/C and 4C and 6C and 9A: the "kDa" is missing from all gels.

Response: Having established in Fig. 2 that SUFU is approximately equal to its calculated molecular mass of 54 kDa, we purposely omitted the molecular weight markers from other figures containing similar gels in order to avoid cluttering up these figures (which are already packed full of information and detail, in our opinion).

Reviewer #3 (Comments to the Authors (Required)):

The major claims are that ERK2 phosphorylates Gli1 and when three serines residues (102, 116 and 130) are all phosphorylated, the affinity of Sufu, the Gli repressor is decreased. Tumour-derived mutations which are located nearby also diminish Sufu binding affinity. This provides a potential mechanism for disease in this case. It has also been demonstrated (by mimetics) that phosphorylation of these residues results in increased Gli transcription factor activity. This is a significant advance in understanding how Hedgehog signalling is activated and can become overactivated in malignant growth.

Response: We are very gratified that Reviewer #3 found our study to be “a significant advance in understanding how Hedgehog signalling is activated and can become overactivated in malignant growth”.

Main points

1) ERK2 phosphorylation of GLI1 modulates GLI1-SUFU binding

Data are strongly supportive. Note, there is a control with GST only shown in the Figures, but I think that the text should mention this to prove that binding is not GST-mediated.

Response: We thank the reviewer for this important comment, which helped us catch a typo in several of the figures in version 1 of our manuscript. In Figures 2, 3 and 6, the lane that was labelled “GST” should have been labelled “0”, i.e. no added GST-GLI1. This ‘0 ng point’ still contained all the other components of the binding assay, including radiolabeled SUFU, binding buffer, bovine serum albumin and glutathione-Sepharose beads (see Materials and Methods). Unfortunately, in Figures 2, 3 and 6, this ‘0 ng point’ was mislabeled as ‘GST’. We apologize for this error, which has been corrected in the figures submitted with this revision.

As the reviewer correctly points out, it is important to prove that binding is not GST mediated, particularly for ³⁵S-SUFU, which we have not previously published on. In the revised manuscript, we show this in Fig S1, which includes samples containing 10 µg GST. In this figure, it is evident that ³⁵S-labeled, *in vitro*-translated SUFU does not bind to GST, even when GST is present at a >20-fold molar excess over the highest concentration of GST-GLI1 used in our experiments (10 µg GST in our standard assay volume of 200 µL is equal to a GST concentration of ~1773 nM; the highest GST-GLI1 concentrations used in our assays is 80 nM).

For Figs 6 and 7, the fact that ³⁵S-labeled, *in vitro*-translated GLI1 does not bind appreciably to GST was established in our previous study (Whisenant et al., 2010). In the Whisenant study, 40 µg of GST was used per control sample, a greater than 80-fold molar excess over the highest concentration of GST-GLI1 used in the experiments shown in Figs 6 and 7.

In order to communicate these important controls to the reader, we added the following text to the end of the Figure 2C legend: “SUFU did not exhibit detectable binding to GST alone, even at a GST concentration of >1.7 µM (Fig S1).” Also, to the end of the Figure 6A legend we added: “Radiolabelled GLI1 does not bind appreciably to GST alone, even when 40 µg of GST is used (Whisenant et al., 2010).”

*2) Phosphorylation occurs on canonical sites and is necessary for binding modulation
Data are strongly supportive.*

Response: We thank the reviewer for this positive comment.

*3) Ser102, Ser116 and Ser130 are efficiently phosphorylated by ERK2.
Data are strongly supportive. However, phosphorylation of S201 is similar to S102. I would like
an explanation of why S201 wasn't pursued in the Sufu release studies.*

Response: In Figure 5, we found that GLI1^{AAA}, a variant with S102, S116 and S130 mutated to alanine, was completely unresponsive to ERK2 treatment. From this, we concluded that “phosphorylation of one or more of the residues S102, S116 and/or S130 is necessary for SUFU release. In addition, since S70, S86, S146 and S201 remained intact in this mutant, this result also demonstrates that phosphorylation of these four residues plays little or no role in promoting SUFU release.”

Furthermore, in Fig 5, we also showed that GLI1^{SSS}, a variant with S102, S116 and S130 intact but with S70, S86, S146 and S201 mutated to alanine, was at least as responsive to ERK2 treatment as was wild-type GLI1. From this we concluded that “phosphorylation on one or more of the residues S102, S116 and/or S130 is sufficient for SUFU release. In addition, since serines 70, 86, 146 and 201 were mutated to alanine in this mutant, this result also demonstrates that phosphorylation of one or more of these four residues is not required for SUFU release.”

That said, we are sensitive to the fact that terms like “necessary and sufficient” may not completely capture the reality of a situation where several different residues contribute to a functional outcome. Hence, the reviewer correctly raises the possibility that phosphorylation of S201 may contribute to SUFU release in a cooperative manner.

In order to examine this possibility, we constructed GLI1^{SSSS}, a new variant of GST-GLI1₆₈₋₂₃₂ in which S102, S116, S130 and S201 are intact, but S70, S86, and S146 are mutated to alanine. We then tested this mutant for SUFU binding following treatment with ERK2 or mock treatment. As shown in the new Figure S4, GLI1^{SSSS} was essentially indistinguishable from wild-type GLI1, and did not display a level of SUFU release greater than wild-type GLI1 or GLI1^{SSS}. To communicate these new results to the reader, we added the following paragraph to the end of the section of the manuscript entitled “Phosphorylation of S102, S116 and/or S130 are necessary and sufficient for SUFU release”:

Although the above experiments seemed to rule out a major role for S201 in SUFU release, we noted that the efficiency of phosphorylation of S201 seen in Fig 4 was almost indistinguishable from S102. Thus, we wondered if phosphorylation of S201 might potentiate the level of release seen with the GLI1^{SSS} variant. To address this possibility, we constructed an additional variant designated GLI1^{SSSS}. In GLI1^{SSSS}, S102, S116, S130 and S201 are intact, while S70, S86, and S146 are mutated to alanine. As shown in Fig S4, GLI1^{SSSS} did not exhibit a greater level of SUFU release than GLI1^{SSS}. This result confirms that phosphorylation of S201 does not play a major role in SUFU release.

4) Phosphorylation of S102, S116 and/or S130 are necessary and sufficient for SUFU release, residues act cooperatively and have the same effect in full-length Gli. Mutations found in tumours reduce binding to Sufu.

Data are supportive. With regards to all mutant experiments, I would like to see data that the mutant proteins are as well expressed as WT and/or that the same amount of protein was used in binding experiments. This could be a supplementary figure.

Response: We thank the reviewer for this suggestion, and have included multiple supplemental figures in the revised manuscript in order to comply with this request.

In all of the bindings assays in our manuscript followed the same general strategy: ³⁵S-labelled protein produced in a rabbit reticulocyte lysate-based cell-free translation system is mixed with a GST-fusion protein purified from *E. coli*.

In Figs 2, 3, 4 and 5, each GST-GLI1 variant was individually purified from bacteria by affinity chromatography, eluted from the matrix, and then carefully quantified by SDS-PAGE and Bradford assay. Based on this quantification, we pipetted equal nanograms of protein into each “master mix”, which was then phosphorylated or mock-phosphorylated as described in Materials and Methods. Following phosphorylation or mock treatment, these mixes were then diluted by serial dilution to obtain the concentration curve shown in the figures. Following SDS-PAGE of the binding assay points, the gels were stained with a Coomassie-blue-based stain in order to visualize the recovered GST-GLI1 protein prior to Phosphorimager analysis and exposure to x-ray film. For the experiments shown in Figures 2, 3, and 5, we have now included representative examples of such Coomassie-stained gels in Supplemental Figures S1, S2 and S3.

In the experiments shown in Figs 6 and 7, the mutants were constructed in the ³⁵S-labelled protein. In these cases, we added the same amount of DNA to the *in vitro*-transcription/translation (TNT) reaction, and then verified that the translation efficiency of wild-type and all mutant proteins were approximately the same by SDS-PAGE analysis followed by Phosphorimager analysis. Any minor differences in translation efficiency between the mutants were adjusted for by including equal numbers of radioactive counts in each binding assay point.

It should be noted that the ³⁵S labelling occurs by the incorporation of trace amount of ³⁵S-methionine into the protein of interest during the TNT reaction. Since the different variants did not differ in the number of encoded methionines (with one exception: GLI1-232 L88M has 10 methionines rather than the wild-type 9), the incorporation of label is proportional to the amount of protein produced. As stated above, any minor differences in translation efficiency between the mutants were adjusted for by including equal numbers of radioactive counts in each binding assay point. To summarize, we are confident that equal amounts of *in vitro*-translated proteins were included in each binding reaction point.

In addition, we have included representative examples of Coomassie-blue stained gels from Figure 6 as Supplemental Figure S5, in order to demonstrate equal recovery of the GST-SUFU protein used in these experiments.

5) Phosphorylation sites in Gli affect transcriptional activity.

Data is strongly supportive.

Response: We thank the reviewer for this positive comment.

Additional Points

Scheidt et al. 2020 performed an interesting phosphoproteomic study which should be referenced and discussed. Their data seem to demonstrate an opposite effect to this manuscript, that ERK/MAPK signaling increases with inhibition of the Hh signalling .

Response: We thank the reviewer for this suggestion. In the revised manuscript, we now include a discussion of the Scheidt *et al.* paper in the Discussion section of our manuscript in the sub-section entitled “Multiple mechanisms of MAPK-mediated crosstalk”. The new text, below, follows the sentence “It seems likely that different mechanisms of crosstalk will predominate to a greater or lesser extent depending on the cell type and context, including the oncogenic load (Aberger & Ruiz i Altaba, 2014).”

New text:

In this regard, it is worth noting that a recent phosphoproteomic study of short-term changes in a HH-responsive human medulloblastoma cell line found that treatment with a HH agonist led to inhibition of ERK signaling, whereas treatment with vismodegib led activation of ERK signaling (Scheidt *et al.*, 2020). The latter finding suggests how ERK-mediated crosstalk could initially maintain some GLI activity in the presence of vismodegib.

Another interesting point for discussion is that in the predicted GLI1 structure produced by AlphaFold, residues 95-111 are predicted to be alpha-helical. One of the phosphorylated residues in this manuscript is contained within this helix. It would be interesting to discuss whether the presence of a helix here would effect phosphorylation and Sufu binding/release.

Response: We thank the Reviewer for sharing this insight. Encouraged by the Reviewer, we have added the following paragraph to the Discussion section of the revised manuscript:

New text:

The region from 1-232 in human GLI1 is not predicted to be intrinsically disordered by tools such as MobiDB (Piovesan *et al.*, 2021). Nevertheless, the structure of the majority of this region cannot be predicted with confidence by Alphafold2 (Jumper *et al.*, 2021; Varadi *et al.*, 2022). Interestingly, the exception is a short region from L95 to N111, which is predicted by Alphafold2 to form two short antiparallel alpha helices connected by a beta turn, with SP102 at the apex of this turn. The same structure is also predicted for the corresponding regions of human GLI2 and GLI3, consistent with the 79% sequence similarity between the three GLI proteins in this region. While it is not immediately apparent how phosphorylation of S102 might alter this structure, it would clearly introduce a negatively a charged phosphate in the vicinity of R100 and S105, two other residues that we showed are critical for SUFU binding.

Again, we thank all three reviewers for their useful and thoughtful comments.

June 3, 2022

RE: Life Science Alliance Manuscript #LSA-2021-01353-TR

Dr. Lee Bardwell
University of California, Irvine
Developmental & Cell Biology
2208 Natural Sciences I
University of California
Irvine, CA 92697-2300

Dear Dr. Bardwell,

Thank you for submitting your revised manuscript entitled "ERK2 MAP kinase regulates SUFU binding by multisite phosphorylation of GLI1". We would be happy to publish your paper in Life Science Alliance pending final revisions necessary to meet our formatting guidelines.

- please add callouts for Figure 2D, Figure 5 F-K, Figure 6A-B and for all supplementary figures to your main manuscript text; if you add a figure callout to the text for one panel, please note that you then need to add a callout for each panel of that figure
- please adjust your Figure 5 to either fit on one page or split into 2 separate figures; these will be displayed in-line in the HTML version of your paper, so please provide them as single page files (Figures); we do not have a limit on the number of main figures and these can be split if necessary for space
- please upload your Supplemental Table 1 file in doc or excel file format and add a legend for it to the main manuscript text

Figure check:

- Figure 2B: there is a splice in the first blot. Please provide source data for this figure
- Figure 2C: Please provide source data for this figure as well
- Figure 5B: all blots seem to have irregularities. Please provide source data for this figure
- figure 5B bottom row is a duplicate of Figure S3A 4th row. Please provide source data for S3A
- please expand the Figure Legend for figure 5 panels F-K.

A. FINAL FILES:

B. MANUSCRIPT ORGANIZATION AND FORMATTING:

Sincerely,

Reviewer #1 (Comments to the Authors (Required)):

I am satisfied with the changes and replies that have been made in response to my comments.

Reviewer #2 (Comments to the Authors (Required)):

The authors have made significant efforts to address the reviewer's comments. The data in the revised manuscript is compelling and supports the conclusions drawn.

Reviewer #3 (Comments to the Authors (Required)):

The authors have addressed the reviewers' feedback very thoroughly. I recommend that this paper now be published.

June 21, 2022

RE: Life Science Alliance Manuscript #LSA-2021-01353-TRR

Dr. Lee Bardwell
University of California, Irvine
Developmental & Cell Biology
2208 Natural Sciences I
University of California
Irvine, CA 92697-2300

Dear Dr. Bardwell,

Thank you for submitting your Research Article entitled "ERK2 MAP kinase regulates SUFU binding by multisite phosphorylation of GLI1". It is a pleasure to let you know that your manuscript is now accepted for publication in Life Science Alliance. Congratulations on this interesting work.

DISTRIBUTION OF MATERIALS:

Again, congratulations on a very nice paper. I hope you found the review process to be constructive and are pleased with how the manuscript was handled editorially. We look forward to future exciting submissions from your lab.

Sincerely,
